# A patient-specific lung cancer assembloid model with heterogeneous tumor microenvironments

Yanmei Zhang[1,2,3,4,10], Qifan Hu [5,10], Yuquan Pei[6,10], Hao Luo[1,2,3], Zixuan Wang[1,2,3], Xinxin Xu[7], Qing Zhang[4], Jianli Dai [4], Qianqian Wang[4], Zilian Fan[1,2,3], Yongcong Fang [1,2,3], Min Ye [1,2,3], Binhan Li[1,2,3], Mailin Chen[8], Qi Xue[9], Qingfeng Zheng[9], Shulin Zhang[9], Miao Huang[6], Ting Zhang[1,2,3], Jin Gu [5] ✉ & Zhuo Xiong[1,2,3] ✉

Cancer models play critical roles in basic cancer research and precision medicine. However, current in vitro cancer models are limited by their inability to mimic the three-dimensional architecture and heterogeneous tumor microenvironments (TME) of in vivo tumors. Here, we develop an innovative patient-specific lung cancer assembloid (LCA) model by using droplet micro-fluidic technology based on a microinjection strategy. This method enables precise manipulation of clinical microsamples and rapid generation of LCAs with good intra-batch consistency in size and cell composition by evenly encapsulating patient tumor-derived TME cells and lung cancer organoids inside microgels. LCAs recapitulate the inter- and intratumoral heterogeneity, TME cellular diversity, and genomic and transcriptomic landscape of their parental tumors. LCA model could reconstruct the functional heterogeneity of cancer-associated fibroblasts and reflect the influence of TME on drug responses compared to cancer organoids. Notably, LCAs accurately replicate the clinical outcomes of patients, suggesting the potential of the LCA model to predict personalized treatments. Collectively, our studies provide a valuable method for precisely fabricating cancer assembloids and a promising LCA model for cancer research and personalized medicine.

Lung cancer is the leading cause of cancer deaths, with ~1.8 million deaths worldwide in 2020[1]. Despite the increasing availability of therapeutic strategies, including targeted therapy and immunotherapy, few patients achieve complete remission, and patient responses are highly variable[2]. It has been appreciated that tumor heterogeneity and tumor microenvironments (TMEs) contribute to tumor development and poor outcomes of anticancer treatment[3,4]. The TME consists of extracellular matrix (ECM) and various cellular components, including immune cells and stromal cells. Cancer-associated fibroblasts (CAFs) are major stromal cells in the TME with the ability to drive cancer metastasis and drug resistance and modulate the immune

microenvironment[5]. The TME varies greatly between and within each patient, causes great disease diversity and poses a major challenge for precision therapy and drug development[6]. Hence, reconstructing a cancer model with tumor heterogeneity and a personalized TME in vitro has become a key issue in cancer research and precision medicine.

In vitro cancer models have contributed tremendously to cancer research and anticancer drug development. However, traditional cancer models, including 2D and 3D sphere cultures, lack the heterogeneous cell subtypes and molecular features of parental tumors[7,8]. Patient-derived cancer organoids are currently the "star"

cancer model that can replicate the pathological morphology and some genetic features of parental tumors. However, conventional cancer organoid models based on matrigel mainly represent tumor epithelium, endogenous stromal and immune cells are gradually lost over time in culture[7,9–11]. Although some studies reconstituted a part of the TME in organoid culture systems by the air-liquid interface (ALI) method[12] or coculturing organoids with TME cells such as CAFs[13,14] and immune cells[15,16], some other studies developed non-Matrigel-based hydrogel 3D cancer models comprised of heterogeneous patient-derived tumor cells and stromal cells[17,18], the models lacked precise controllability and uniformity in addition to labor costs. Some other cancer organoids derived from minced tumor fragments could maintain the native tissue architecture and TME cell components. However, manual tissue mincing results in non-reproducible fragment sizes and nonuniform environments[9,19,20]. On the other hand, a limited number of millimeter-scale tumor fragments derived from small tumor tissues are limited in application in high-throughput drug screening.

Assembloids are 3D structures formed from the fusion and functional integration of multiple cell types or organoids, which are the latest tools for understanding human development and disease and are now considered at the leading edge of stem cell research[21–24]. Bladder tumor assembloids were created and partially recapitulated the in vivo pathophysiological features of urothelial carcinoma[23,25].

Currently, assembloids are mainly fabricated by coculture[26,27] and 3D extrusion printing methods[25]. The morphology and structure of assembloids fabricated by coculture methods are difficult to control and have poor intrabatch consistency[28]. Although a kidney organoid model[29] with tissue morphology could be fabricated by using extrusion-based 3D printing method, only 18 organoids with diameter of ~2 mm (0.55 μL in volume for each organoid), could be generated per minute. 3D extrusion bioprinting is limited in generating micron-size tissue models rapidly with requisite size and accuracy[29–32]. Rapid preparation of tumor assembloids with good intrabatch uniformity remains a great challenge.

In this study, we report an innovative patient-specific lung cancer assembloid (LCA) model generated by microinjection-based droplet microfluidic technology that enables precise manipulation of clinical microsamples and high-throughput generation of LCAs (Fig. 1). LCAs are achieved by evenly encapsulating patient tumor-derived TME cells and lung cancer organoids (LCOs) inside gelatin methacryloyl (GelMA)-Matrigel microgels with good cytocompatibility. This LCA model demonstrates good intrabatch consistency in size, cell composition and drug response profiling. In addition, these LCAs represent the TME and tumor heterogeneity of their parental tumors and replicate the clinical responses of patients with lung cancer, highlighting the potential utility of our LCA model for basic research and personalized drug screening.

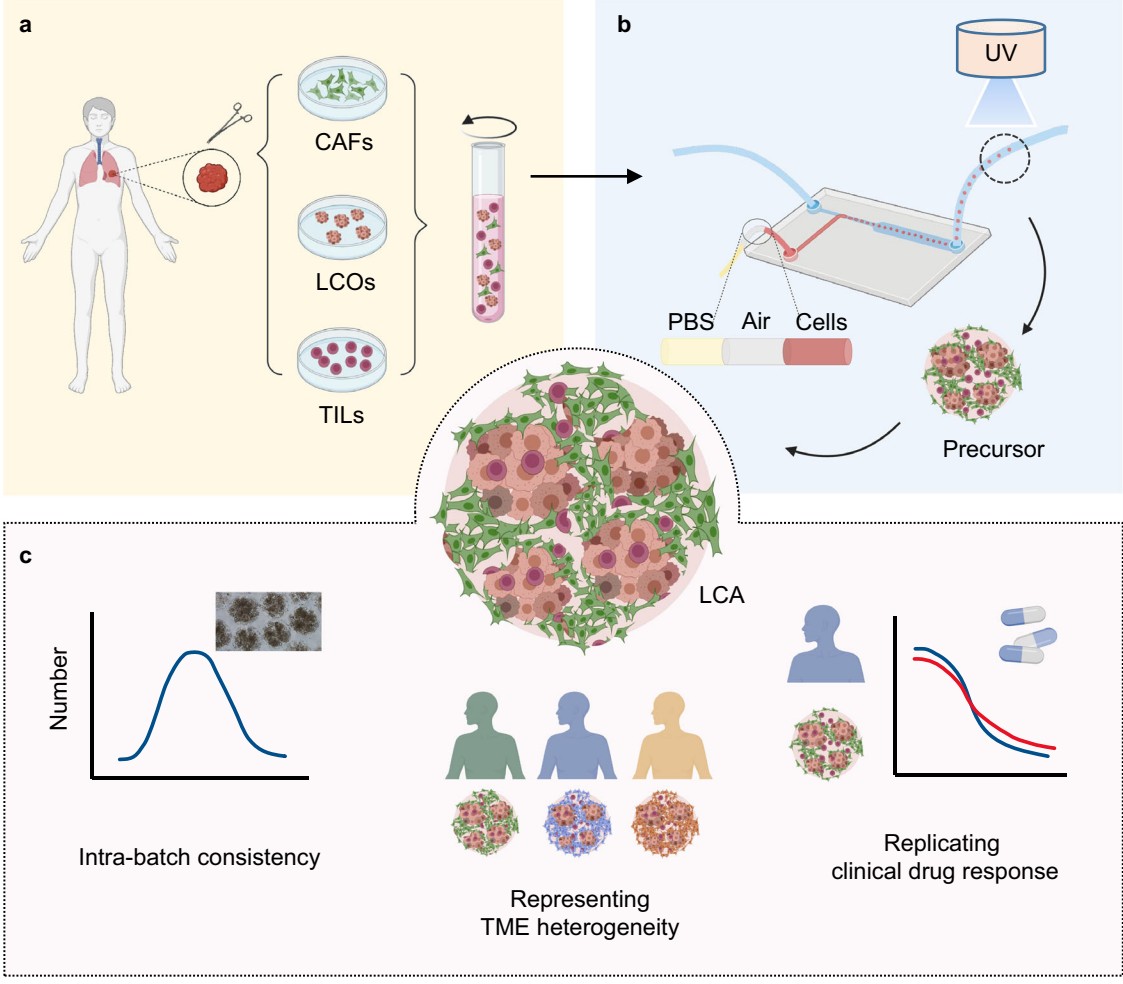

**Fig. 1 | The patient-specific lung cancer assembloid (LCA) model. a** The schematic illustration of the preparation of bioinks loaded with lung cancer organoids and TME cells. **b** Fabrication of LCAs by using a droplet microfluidic technology based on a microinjection strategy. **c** The advantages of LCAs which show good intra-batch consistency, represent TME heterogeneity of parental tumors and replicate clinical drug responses. LCOs, lung cancer organoids; CAFs, cancer associated fibroblasts; TME, tumor environment; TILs, tumor infiltrating lymphocyte cells.

## Results

### Establishment of uniform LCAs through a microinjection strategy-based droplet microfluidic technology

The rise of cancer assembloids provides a promising tool for cancer research. To generate uniform LCAs with personalized TMEs in a high-throughput manner, we developed an LCA platform using innovative microinjection strategy-based microfluidic technology (Supplementary Fig. 1). The platform is designed for cell-laden GelMA-Matrigel manipulation, mainly consisting of bioink preparation and LCA generation processes (Fig. 1a, b). A total of 49 clinical tumor samples were collected, and tumor-derived cells (LCOs and TME cells) from 36 patients were used to fabricate LCAs by using droplet microfluidics. We successfully fabricated LCAs in 35 patients with a 97.2% success rate. (Supplementary Fig. 2a-g, Supplementary Table 2). LCOs and TME cells were encapsulated into the optimized GelMA-Matrigel hydrogel for bioink preparation at a density of $4 \times 10^7\,\mathrm{mL}^{-1}$ cells. The microinjection module is designed for microsample manipulation and includes a silica tube with an inner diameter of 500 μm for sucking bioink into the tube. The bioink is separated by air from the booster reagent PBS to prevent it from being diluted. The end of the tube could be attached to the forming module, a T-junction chip where the cell-laden hydrogel is subsequently sheared by mineral oil into monodisperse droplets (Fig. 1b, Supplementary Fig. 1). The flow rates of the bioink and oil phase were optimized at 1 and 5 mL h$^{-1}$, respectively, to form uniform assembloid precursors with a size of 400–500 μm encapsulating a certain number of cells (e.g., 1500–2500 cells). The droplets are subsequently UV photo-crosslinked with controllable UV intensity and form stable cell-laden microgels (LCA precursors) that can grow into LCAs with patient-specific TMEs after 3 days of culture (Fig. 2a–c).

To generate LCAs with good mechanical properties and biological activity, we chose GelMA and Matrigel composite hydrogels because GelMA hydrogels are widely used for their excellent processing capability, tunable mechanical properties, and biocompatibility even for immune cells[16,33–35], while Matrigel hydrogels can provide a favorable tumor microenvironment for patient-derived organoids[36,37]. The material ratios were optimized to ensure good biocompatibility and formability. The analysis of compressive mechanical property indicated that the hydrogel consisting of 15% (v/v) Matrigel and 6% (w/v) GelMA (termed 6–15) exposed to 90 mW of blue light at 405 nm for 40 s showed comparable mechanical properties to those of patient lung tumors (19.1 ± 0.4 vs. 27.9 ± 3.9 kPa) (Fig. 2d, Supplementary Fig. 3a). More importantly, the 6–15 hydrogel maintained good cell viability and cell proliferation compared with the 6–0 (6% GelMA) and 6–30 (6% GelMA plus 30% Matrigel) hydrogels (Fig. 2e–g, Supplementary Fig. 3b–d).

In addition, uniform LCA precursors with good intrabatch consistency in terms of size, cell composition and distribution could be generated using 6–15 hydrogels in this platform (Fig. 2h, Supplementary Fig. 3e, f). Even 10 μL hydrogels containing $10^6$ - $10^8$ cells mL$^{-1}$ could be successfully manipulated to generate ~200 uniform LCAs with sizes of 400–500 μm (~0.05 μL per LCA) within 1 min (Fig. 2i, Supplementary Fig. 3g, h). Encouraged by the results, we successfully generated uniform LCAs directly using the limited number of cells derived from tiny tumor needle biopsies (Fig. 2j, Supplementary Fig. 3h). This suggested broad application of this platform in the rapid fabrication of cancer assembloid models even using microsamples such as biopsies that can be easily obtained from patients with intermediate and advanced tumor stages[38].

The ability to bank such assembloids will improve the utilization of LCAs and provide researchers with the opportunity to generate living biobanks, which will substantially contribute to basic and translational research in a wide range of areas[39]. Therefore, we performed a thawing test for cryopreserved LCAs. LCAs could successfully reconstitute their biological properties after being frozen for 2 months. The morphology and diameter of LCAs before freezing and after thawing showed great similarity, and cells assembled inside the LCAs maintained good viability and proliferation after thawing (Fig. 2k, l), suggesting that the LCAs could be cryopreserved as a biobank for further applications.

### LCAs maintain the heterogeneous histology and TME features of parental tumors

Tumors have the features of inter- and intratumor heterogeneity, including but not limited to cellular and histological heterogeneity[40,41]. The LCAs derived from the same patient or different patients showed heterogeneous morphology, suggesting the maintenance of inter- and intratumor heterogeneity of patients (Supplementary Fig. 4a). To further assess whether the LCAs resemble their corresponding parental tumors at the histological level, we performed histological analyses. The hematoxylin and eosin (H&E) staining results showed that the LCAs had similar histological features to their parental tumors (Fig. 3a, Supplementary Data 1). Stromal cells (red arrows) were observed wrapping around the tumor cells (black arrows) and forming junctions with each other as indicated by the arrows. LCAs derived from adenocarcinomas (ACs) of different patients maintained the intertumoral heterogeneity of cancer cell differentiation degree, and the expression patterns of EpCAM, cytokeratin 7 (CK7) and Ki67 were also retained in the LCAs (Fig. 3b, Supplementary Fig. 4a, b). It is worth noting that LCAs replicated the heterogeneous expression of CK7 and Ki67 markers within LC14 tumors, which indicated that LCAs also recapitulated the intratumoral heterogeneity of the original tumor tissues.

To further characterize the TME in LCAs, we performed immunohistological analysis using specific markers of TME cells. We observed heterogeneous cells within the LCAs, including the EpCAM$^+$ tumor cells, α-SMA$^+$ CAFs and CD45$^+$/CD3$^+$ immune cells (Fig. 3c, d, Supplementary Fig. 4c–e), which indicated that LCAs could recapitulate a certain tumor microenvironment of the parental tumors. In particular, LCAs of 5 patients generated by droplet microfluidics demonstrated a more uniform shape, cell distribution and TME maintenance than those generated by coculture in U-bottomed ultra-low attachment microplates (ULAs), which are often used to generate 3D spheroid tumor models[42] (Supplementary Fig. 4f, g). Ki67 and hypoxia probe staining showed that tumor cells proliferated well without an obvious tumor hypoxia zone within the LCAs (Supplementary Fig. 4h, i). The small size of our LCAs (400-500 μm) may contribute to oxygen and nutrient delivery, and we could mimic tumor hypoxia gradients by increasing the size of LCAs according to the need for research[20].

Overall, these results demonstrated that our LCAs could maintain the TME and heterogeneous histology of corresponding parental tumors.

### LCAs maintain the transcriptomic and genomic signatures of parental tumors

To determine whether LCAs maintained the transcriptomic landscape of their corresponding parental tumors. We performed bulk RNA sequencing (RNA-seq) on samples from 4 patients (LC14, LC28, LC51 and LC52), including tumor samples, matched adjacent normal lung tissues for references, and the corresponding LCAs derived from the tumor organoids cultured for 3 weeks (LC28 and LC51) to 12 weeks (LC14). Transcriptome-wide comparisons demonstrated the high similarity between LCAs and their corresponding parental tumors, with an overall correlation coefficient of 0.86. However, the similarity decreased with the culture time of both cancer organoids and TME cells (the correlation coefficients of gene expression in LC28 and LC14 were 0.9 and 0.75, respectively) (Fig. 4a, Supplementary Fig. 5a, b), consistent with that in organoids[43].

Pathway enrichment analysis with the upregulated genes in LCAs (vs. normal) and parental tumors (vs. normal) was performed among

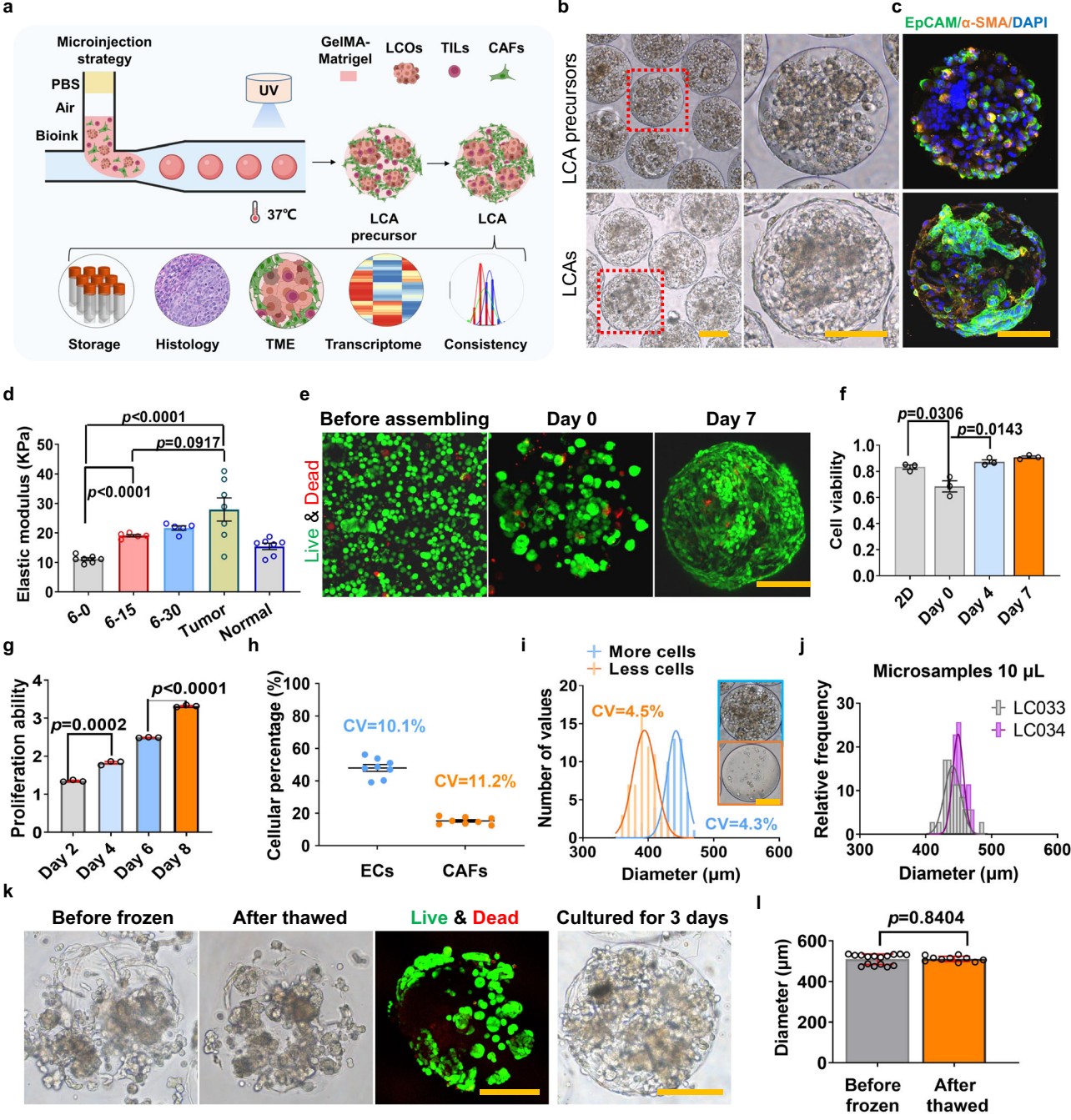

**Fig. 2 | Establishment of uniform LCAs through a microinjection strategy-based droplet microfluidic technology. a** Schematic representation of LCA fabrication strategy and characterization. **b** Representative bright field microscopy images of LCAs (LC05) at day 0 (LCA precursor) and day 3 post fabrication. The right images are the enlarged views circled by the red lines. The experiment was repeated in 35 patient samples. **c** Immunofluorescence staining of the EpCAM and α-SMA markers in LCA precursors and LCAs. The experiment was repeated in 15 patient samples. **d** Compressive elastic modulus of GelMA-Matrigel hydrogels, lung cancer tumors and the matched adjacent normal lung tissues (*n* = 7 independent samples for 6–0, tumor and normal groups, *n* = 5 independent samples for 6–15 and 6–30 groups). **e** Representative live (green) & dead (red) staining images of LCAs at day 0 and day 7 post fabrication. The experiments are repeated in LCAs of 6 patient samples. **f** The normalized cell ability of cells before assembling and assembled as LCAs at day 0

and day 7 (*n* = 3 biologically independent samples). **g** Quantitative analysis of cell proliferation ability of LCAs over culture time (*n* = 3 biologically independent cells). **h** The cellular percentages of LCOs and CAFs in intra-batch LCAs (*n* = 8 independent LCAs). **i** Histograms of LCA size distribution for high cell density (more cells, $10^8$ mL$^{-1}$, *n* = 53 LCAs) and low cell density (less cells, $10^6$ mL$^{-1}$, *n* = 62 LCAs). **j** Histograms of LCA size distribution fabricated with 10 μL of microsamples derived from tumor biopsies (LC33 & LC34) (*n* = 35 independent LCAs). **k** Representative bright-field images of LCAs before freezing and after thawing. The experiments are repeated in LCAs of 6 patient samples. **l** Diameters of LCAs before freezing (*n* = 16 independent LCAs) and after thawing (*n* = 10 independent LCAs). Scale bar, yellow bar, 200 μm; two-sided student's *t* test is used, data are presented as mean ± S.E.M. Source data are provided as a Source Data file.

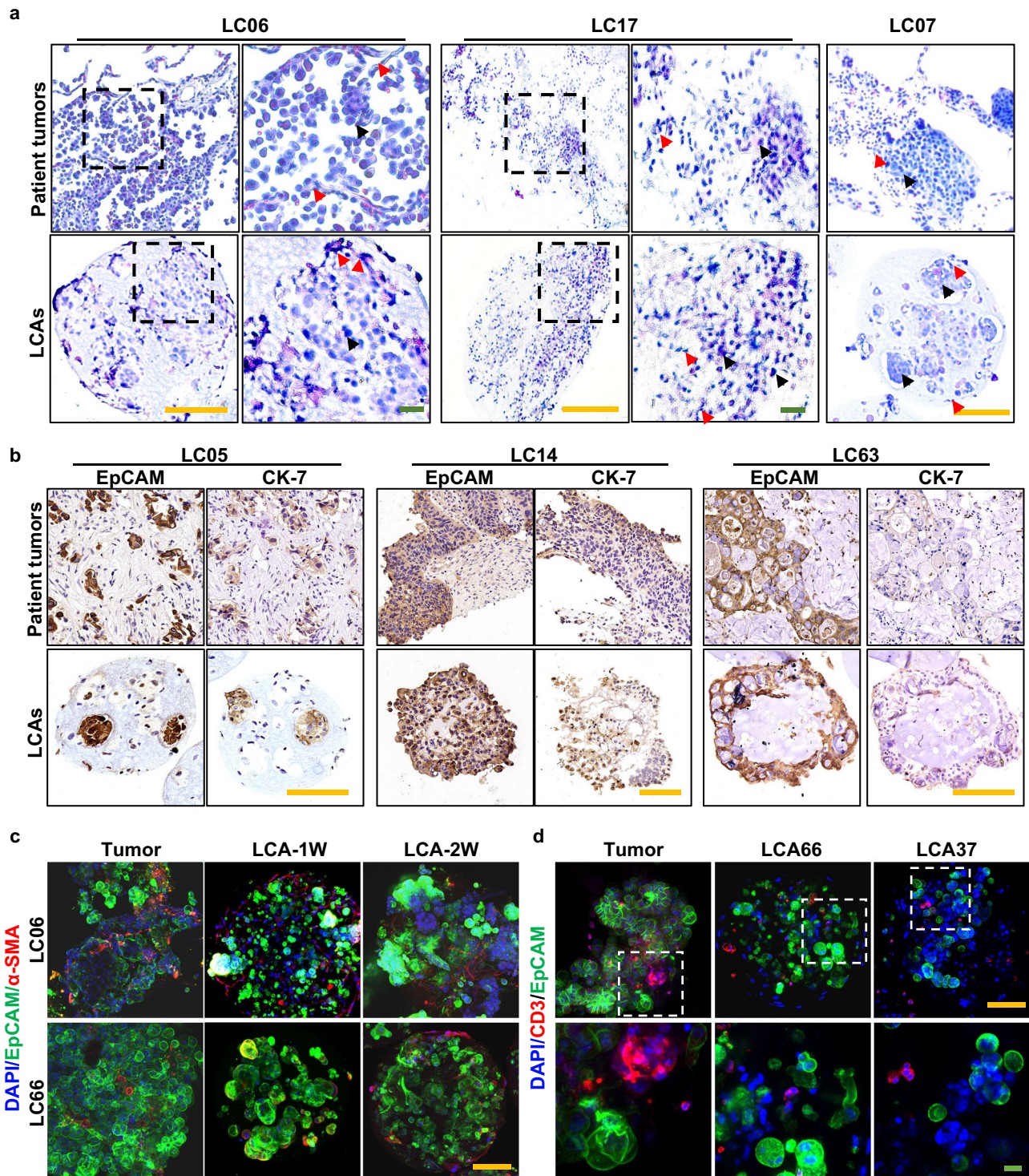

**Fig. 3 | LCAs maintain the heterogeneous histology and TME features of parental tumors. a** H&E staining images of LCAs and their corresponding parental tumors (LC06, LC17 and LC07). Black arrows indicated the tumor cells and the red arrows indicated the stromal cells. **b** Concordant expression of CK7 and EpCAM in parental tumors (LC05, LC14 and LC63) and their derived LCAs. **c** Immunofluorescence staining of human α-SMA and EpCAM in tumor fragments and corresponding LCAs cultured for 1 week and 2 weeks. **d** Immunofluorescence staining of human CD3 and EpCAM in tumor fragments and LCAs (LCA37 and LCA66). For (**a**–**d**) each experiment was repeated independently with similar results for 3 times. Scale bar, yellow bar, 200 μm, green bar, 20 μm.

all the parental tumors and corresponding LCAs. The results demonstrated that the top 20 enriched pathways in parental tumors or LCAs were also highly enriched in corresponding LCAs or parental tumors. The shared enriched pathways of upregulated genes of the four LCA-parental tumor pairs were mainly involved in biological processes such as the mitotic cell cycle, cell proliferation, epithelial cell differentiation,

and extracellular matrix organization (Fig. 4b, Supplementary Fig. 5c). The shared enriched pathways of downregulated genes were mainly associated with the ECM and circulatory system (Fig. 4c), which are some of the main differentiating features between tumors and corresponding normal tissues[44–46]. A systematic comparison between LCAs and parental tumors showed that the major differences were the

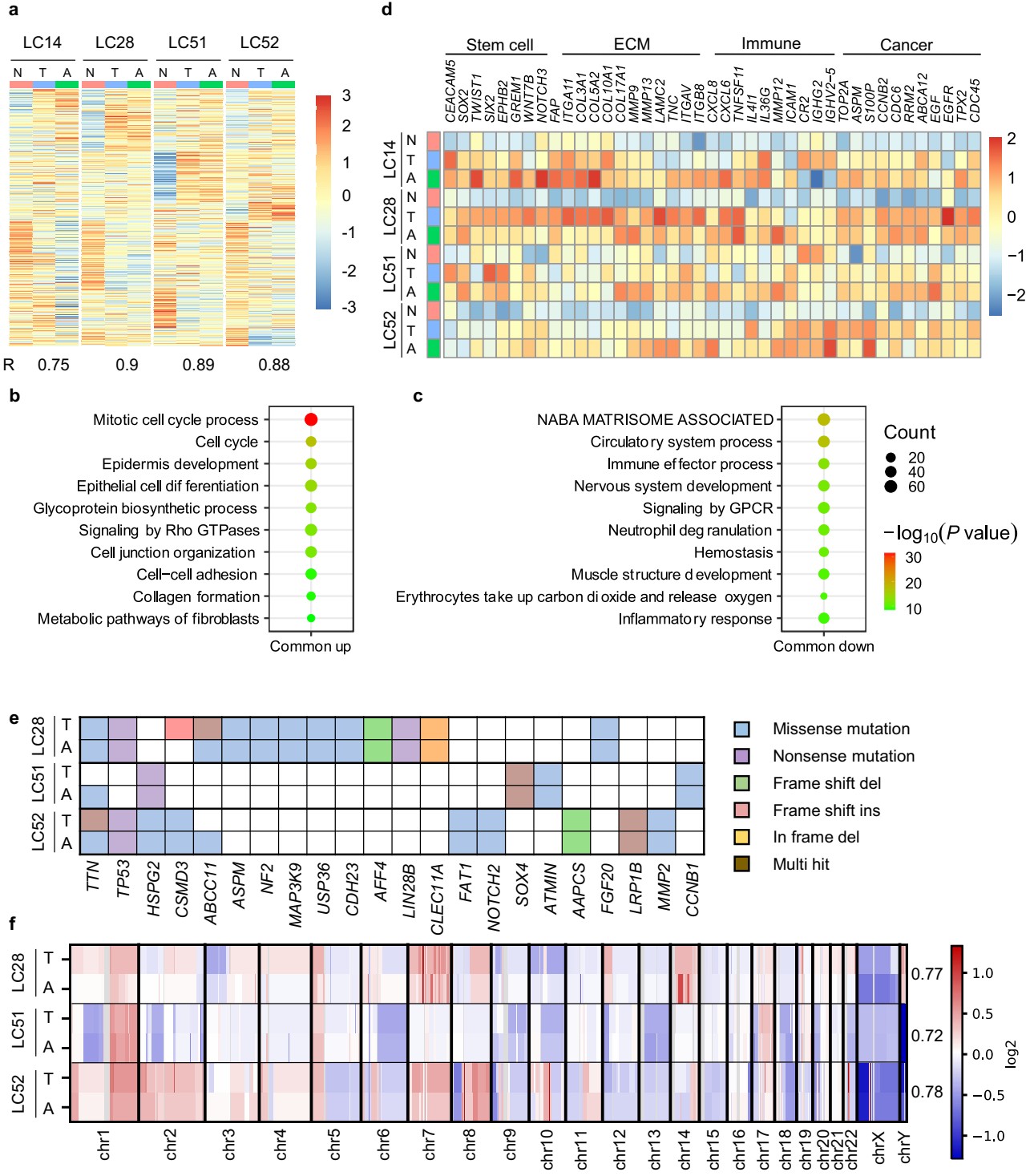

**Fig. 4 | LCAs maintain the transcriptomic and genomic signatures of parental tumors. a** Gene expression heatmap of 6577 differentially expressed genes (see Methods) in normal tissues, parental tumors and LCAs of four patients (LC14, LC28, LC51 and LC52). N, normal tissue; T, tumor; A, LCAs. Pearson correlation coefficients of log(CPM) values of all genes between tumor and LCA from each patient were shown at the bottom. **b, c** Bubble plots of enriched gene ontology biological process terms in the shared upregulated and downregulated genes of the LCAs and parental tumors compared to the matched normal tissues. The *p*-values are derived by Metascape which utilizes the well-adopted hypergeometric test and Benjamini-Hochberg *p*-value correction algorithm. **d** Heatmap of the enriched maker genes of different cell populations (cancer stem cells, immune cells and lung cancer cells) between tumors and corresponding LCAs. **e** Somatic mutations found in lung cancer-related genes are present and conserved between LCAs and parental tumors. Mutation types are indicated in the legend. **f** Heat map illustrating genome-wide copy number variations (CNVs) of lung tumor−LCA pairs of three patients (LC28, LC51 and LC52). Numbers on the right side indicate the Pearson correlation coefficient for each pair of samples. DNA copy number gains (red) and losses (blue) found in the original lung cancer tissues are conserved in the corresponding LCAs.

upregulation of the genes involved in bioprocesses such as epithelial cell proliferation, inflammatory response and wounding response and the downregulation of genes associated with circulatory system processes and ECM organization (Supplementary Fig. 5d). These different enrichment pathways likely suggested that LCAs maintained the immune microenvironment but relatively lacked vasculature. Furthermore, marker gene expression analysis indicated that LCAs and their original tumors showed similar expression profiles of cancer stem cell-related genes (e.g., *SOX2, SIX2, EPHB2*), CAF-related genes (*FAP*), ECM-related genes (e.g., *COL3A1, COL5A2, COL10A, VCAN*), immune cell-related genes (e.g., *CXCL8, CXCL6, TNFSF11, IL4I1*) and cancer cell-related genes (e.g., *TOP2A, ASPM, S100P, RRM2*) (Fig. 4d), indicating that LCAs largely maintained the cellular heterogeneity of parental tumors.

To determine whether LCAs retained the genomic alterations of their parental tumors, we performed whole-exome sequencing analysis of tumors, matched normal tissues and LCAs of 3 patients (LC28, LC51 and LC52). Most somatic variants identified in the parental tumors were found in corresponding LCAs (Supplementary Fig. 5e). The discovered genomic alterations agreed with previously reported mutations typical of lung cancer, such as *TP53, TTN, MAPK3 and ABCC11* (Fig. 4e). Copy number variants (CNVs) in the parental tumor were also detected in the corresponding LCA at similar copy number ratios (Fig. 4f, Supplementary Fig. 5f). Overall, these results demonstrated that LCAs retain the transcriptomic and genomic characteristics of their corresponding parental tumors.

## LCAs maintain the cell-type heterogeneity and cell–cell interaction signatures of parental tumors

To further investigate cell-type heterogeneity and cell–cell interaction signatures, we performed single-cell RNA sequencing (scRNA-seq) analysis of parental tumors from 2 patients and corresponding LCAs cultured for 1 and 2 weeks. We first integrated all cells from six datasets and identified six major cell types, including epithelial cells, fibroblasts, T cells, B cells, mast cells and macrophages (Fig. 5a, see Method section). Based on the markers[47,48] shown in Supplementary Fig. 6a. The epithelial cells could be further clustered into 6 subtypes (e.g., epithelial basal cells, AT2-like cells, AT2 cells, club cells and cells in a proliferating state), and the fibroblasts and T cells were both clustered into 2 subtypes (nonproliferating and proliferating cells), suggesting that tumor and TME heterogeneity existed in these samples (Fig. 5b). LC55 and LC66 showed cell clustering similarity in shared major cell types except epithelial cells (basal and club cells were enriched in LC55, while AT2-like and AT2 cells were enriched in LC66) (Fig. 5c, Supplementary Fig. 6b), suggesting great patient heterogeneity in epithelial cells. Notably, all the cell types of patient tumors were maintained well in the corresponding LCAs, and the proportions of cell types in 1 week-old LCAs were similar to those in parental tumors, whereas those in 2 week-old LCAs showed a lower degree of similarity to parental tumors (Fig. 5d, Supplementary Fig. 6c). Further investigation of transcriptome similarities between LCAs and their parental tumors showed that 1 week-old and 2 week-old LCAs displayed ~88% and 84% overall similarity to parental tumors, respectively. The epithelial and fibroblasts of LCAs showed a slight decrease in overall similarity (92% to 86% in epithelial cells; 91% to 87% in fibroblasts) to those of parental tumors over time. Of note, T cells of both 1 week-old and 2 week-old LCAs displayed high overall similarity to parental tumors with 93% and 92% similarity, respectively (Fig. 5e, Supplementary Fig. 6d).

To further investigate the maintenance of cell–cell communications in LCAs, ligand–receptor interaction analysis across all major cell types was performed. The results showed that ligand–receptor interactions were exhibited among major cell types with large cell populations (epithelial cells, fibroblasts, T cells and B cells) in all samples. Both 1-week-old LCAs and 2 week-old LCAs maintained these cell–cell interactions (Fig. 5f, Supplementary Fig. 7a). Moreover, the overall similarity of ligand–receptor interactions of fibroblast-epithelial cells and epithelial-fibroblasts between LCAs and parental tumors was ~78% and 68%, respectively, suggesting the maintenance of communication between tumor cells and CAFs (Fig. 5g). The similarities of ligand–receptor interactions from fibroblasts to epithelial cells did not change significantly over time (76.8% **to** 78.8%), while a slight decrease in similarities was observed in the interactions from epithelial cells to CAFs of the LC55 sample (73% – 63%) (Fig. 5g), consistent with that in LC66. This may be caused by changes in cell proportions and expression levels of ligand or receptor genes over time. Notably, the *ligand–receptor* pairs *HGF-MET* and *FGF-FGFR1* of fibroblast-epithelial cells and T*GFβ-(TGFBR1 + TGFBR2)* of epithelial-fibroblasts were observed in both parental tumors and LCAs (Fig. 5g, Supplementary Fig. 7c), which are associated with the functional heterogeneity of CAFs[49].

Together, single-cell RNA-seq analyses highlight marked cellular heterogeneity and cell–cell communications in LCAs, further supporting that LCAs recapitulate cell-type heterogeneity and molecular properties of corresponding parental tumors.

## LCA model could reconstruct and identify functionally heterogeneous CAFs

CAFs are highly functionally heterogeneous and correlate with clinical patient responses to drug treatment[49]. However, the characterization of the landscape of CAF functions is challenging because of the lack of specific markers and effective models. Most current fibroblast models available for cancer research are based on 2D coculture[49] or 3D coculture systems[50,51], which lack patient-specific tumor cells and matched CAFs, making it difficult to represent tumor pathological features and assess CAF functions. We established a stable biobank of lung cancer organoids and CAFs derived from patient tumors, enabling us to generate LCAs with patient-specific CAFs. Therefore, we wondered whether our patient-specific LCA models could reconstruct and evaluate the heterogeneous functions of CAFs.

We first identified the heterogeneity of a cohort of 7 patient-specific CAFs according to the HGF and FGF7 secretion levels[49], which play a major role in the development of radioresistance, chemoresistance and EGFR-inhibitor resistance[49,52]. We found heterogeneous CAFs, including the subtype with HGF^high FGF7^high/low expression (LC05, LC17, LC22, LC19 and LC27) and the subtype with HGF^low FGF7^low expression (LC23) (Fig. 6a, b). The distribution and proportion of CAFs were heterogeneous in these tumors from different patients (Fig. 6c, Supplementary Fig. 8a). We next generated LCAs with patient-derived LCOs and corresponding CAFs to assess the effect of CAFs on tumor growth and tumor responses to drug treatment. LCOs encapsulated in the microgels were used as the control group (Supplementary Fig. 8b). Encouragingly, we observed heterogeneous effects of CAFs on the matched tumor cells. LC22 CAFs with HGF^high FGF7^high expression significantly promoted the growth of cancer organoids in LCAs (Fig. 6d, e) and improved the resistance of LCAs to chemotherapeutic drug (Taxol) treatment (Fig. 6f). LC17 CAFs also protected the corresponding cancer cells carrying EGFR mutations and contributed to EGFR-targeted drug (osimertinib) resistance in LCAs (Supplementary Fig. 8c, d; Supplementary Table 2), which was consistent with CAF-driven EGFR inhibitor resistance[49,53]. However, LC23 CAFs with HGF^low FGF7^low levels did not facilitate the growth of the matched cancer cells or protect tumor cells from Taxol treatment (Fig. 6d, f). The combined drug treatment assay demonstrated that a CAF-targeted drug (pirfenidone, PFD) significantly enhanced the sensitivity of LC22 cancer cells to Taxol (Fig. 6g) and caused more apoptotic as well as dead EpCAM^+/E-cadherin^+ cancer cells (Fig. 6h, i, Supplementary Fig. 8e). However, LCAs of LC23 showed a very limited response to combined treatments with Taxol and PFD (Fig. 6j, Supplementary Fig. 8f). Moreover, compared to the LCO group, CAF-targeting

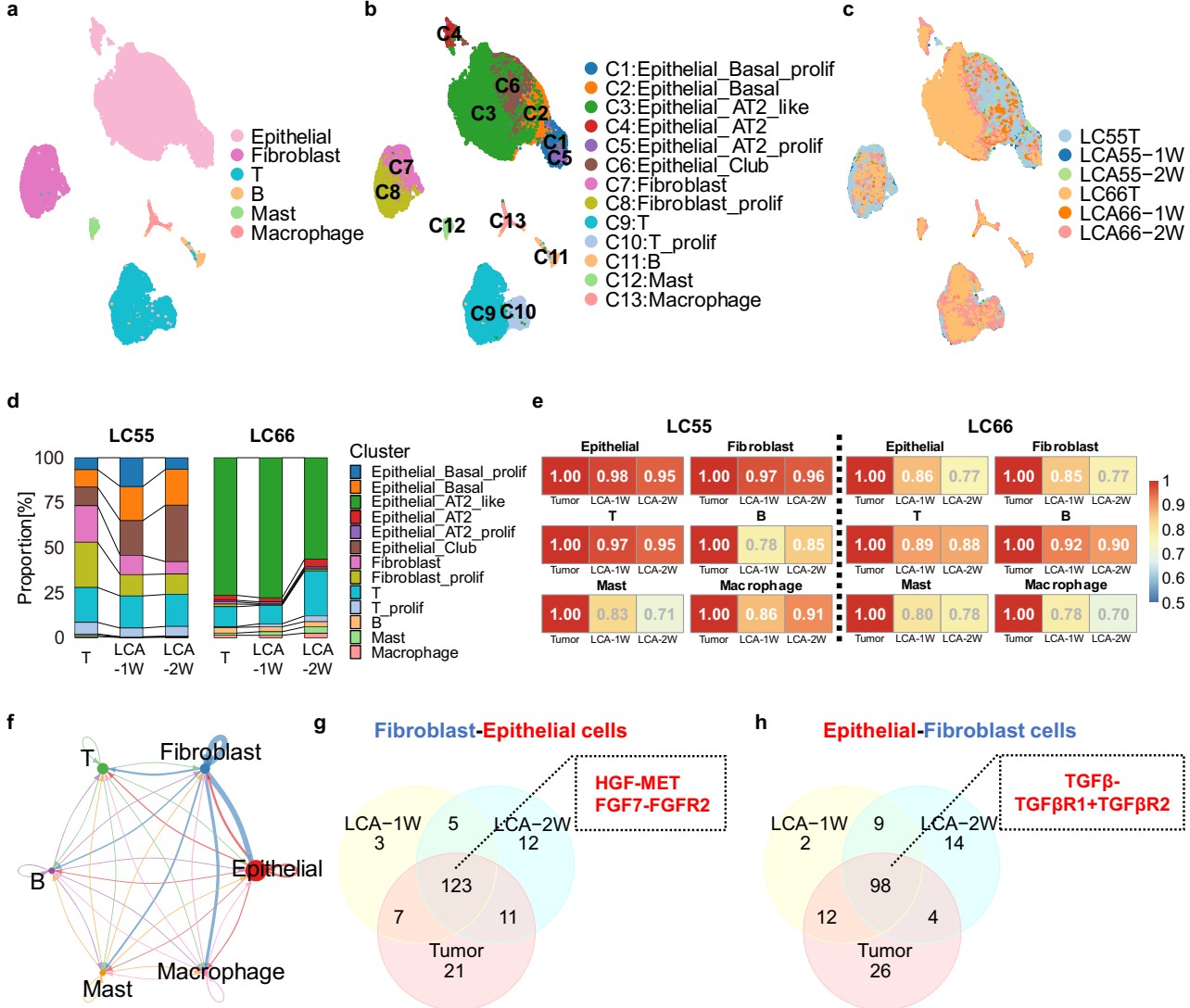

**Fig. 5 | LCAs maintain the cell-type heterogeneity and cell-cell interaction signatures of parental tumors. a** UMAP plot showing the coarsened clustering results colored by 6 major cell types. **b** UMAP plot showing the fine clustering results colored by 13 cell types. **c** UMAP plot showing all cells colored by sample idents. **d** Comparative analysis between primary tumors (LC55 and LC66) and corresponding LCAs for the proportions of individual cell types. T, T cells; B, B cells; Prolif, Proliferation; AT2, Alveolar type 2. **e** Pearson correlation coefficients (PCC) obtained using cell type-specific differentially expressed genes. **f** Circle plot of cell-cell communication network. The width of edges represents the strength of the communication. **g** Venn diagram showing the overlap in ligand-receptor pairs from CAFs to epithelial cells between LCAs and the parental tumor of LC55 patient. The ligand-receptor pairs of interest are listed in box. **h** Venn diagram showing the overlap in ligand-receptor pairs of epithelial-CAFs between the LCAs and the parental tumor of LC55 patient. The ligand-receptor pairs of interest are listed in box.

treatment did not significantly enhance the sensitivity of EGFR-mutated LC23 tumor cells in LCAs to osimertinib (Fig. 6k, l, Supplementary Fig. 8g), further suggesting little protection of LC23 CAFs for the corresponding tumor cells. The mechanism may be due to the limited activation of the paracrine signaling pathways caused by low secretion of HGF and EGF7 from CAFs, and these signaling pathways (e.g., PI3K/Akt, MAPK signaling) contribute to cancer proliferation and antiapoptosis[49,54,55]. Our findings are different from the study that revealed that cancer assembloids exhibited stronger resistance to all chemotherapeutic drugs than conventional tumor organoids[25], which may be due to the functional heterogeneity of CAFs, presenting distinct protection for cancer cells in our patient-specific LCA models[49,56,57].

In conclusion, these data indicated that heterogeneous CAFs contribute differently to tumor protection, and our LCA models could serve as an effective tool to study the functional heterogeneity of CAFs and predict personalized combination treatments.

## LCAs as a powerful preclinical model for personalized drug testing

Preclinical cancer models with the potential to predict heterogeneous drug responses are urgently needed for precision medicine. Whether LCAs can be used as an effective tool for precision oncology depends on some critical standards including cell viability consistency across testing wells, reflection of heterogeneous drug responses of patients and the consistency between testing results and the clinical responses[11,58].

The reflection of heterogeneous drug responses and LCA cell viability consistency across the wells of parallel experiments were first analyzed. We performed large-scale cell viability tests in LCAs and corresponding LCOs of patients. As indicated by the dose-response curves and LogIC50 values, both LCAs and LCOs could recapitulate the inter- and intra-patient heterogeneity of responses to the chemotherapies commonly used in clinic for lung cancer (Fig. 7a, b, Supplementary Fig. 9a, b). However, there were still significant difference in drug responses between LCA and LCO groups. For example,

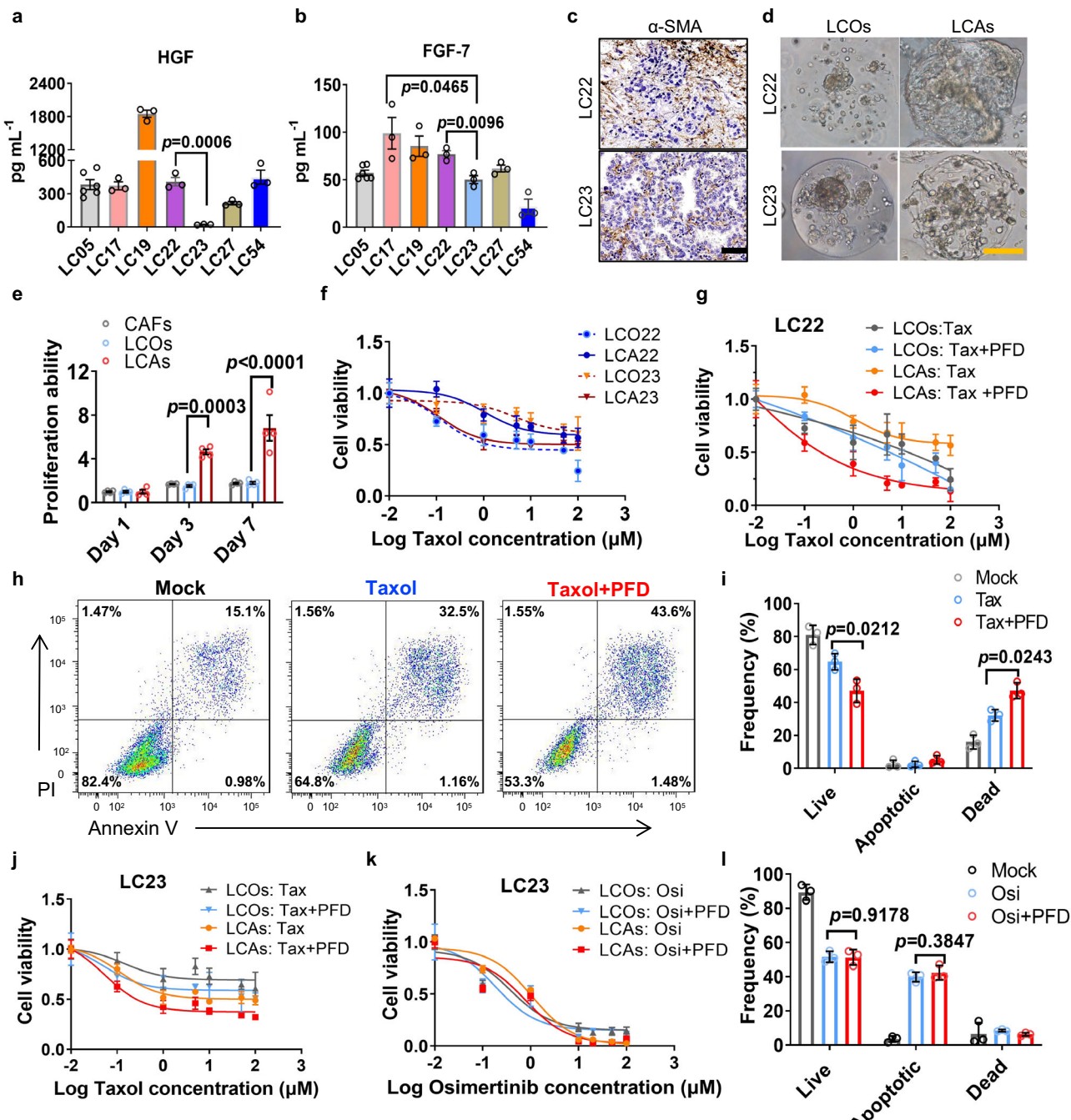

**Fig. 6 | LCAs can reconstruct and identify functionally heterogeneous CAFs.**
**a, b** HGF and FGF7 secretion levels of CAFs derived from tumors of 7 patients via enzyme linked immunosorbent assay (ELISA) ($n = 5$ biologically independent samples for LC05, $n = 3$ biologically independent samples for the other patients).
**c** IHC staining for α-SMA+ CAFs in the tumors derived from LC22 and LC23 patients.
**d** Brightfield images of LC22 and LC23 LCOs, and LCAs encapsulated separately in microgels after 7 days of culture. The experiment was repeated in 7 patient samples. **e** LC22 CAFs promoted the proliferation ability of LC22 assembloids ($n = 4$ biologically independent samples). **f** Dose−response curves of LCOs and LCAs of LC22 and LC23 after 3 days of Taxol treatment ($n = 4$ biologically independent samples). **g** Dose−response curves of LC22 LCOs and LCAs after 3 days of treatment

with Taxol and 2 μM PFD. PFD, Pirfenidone ($n = 4$ biologically independent samples). **h,i** Flow cytometry analysis and quantification of EpCAM+ tumor cell apoptosis of LC22 LCAs with the treatment of 1 μM Taxol and 2 μM PFD ($n = 3$ biologically independent samples). **j** Dose−response curves of LC23 LCAs and LCOs after 3 days of treatment with Taxol and 2 μM PFD ($n = 3$ biologically independent samples). **k** Dose−response curves of LC23 assembloids after 3 days of treatment with Osi and 2 μM PFD. Osi, Osimertinib ($n = 3$ biologically independent samples). **l** Quantification of the live, apoptotic, and dead cells in EpCAM+ tumor cells of LC23 LCAs with the treatment of 1 μM Osi and 2 μM PFD ($n = 3$ biologically independent samples). Scale bar, yellow bar, 200 μm, black bar, 50 μm; two-sided Student's $t$ test is used and data are presented as mean ± S.E.M. Source data are provided as a Source Data file.

LCO36 was sensitive to both PC (Pemetrexed + Carboplatin) and TC (Taxol + Carboplatin), whereas LCA36 was resistant to PC. Additionally, responses of most LCAs to the chemotherapeutic and targeted drugs were decreased compared with LCOs (Fig. 7b, Supplementary Fig. 9b), similar with the previous study[25]. These differences may be

caused by the tumor microenvironments existed in LCAs, which may reflect the real responses of tumors with microenvironment to drugs[59–61].

Then, the cell viability consistency across parallel experiments was assessed. the drug response heatmap showed a relatively

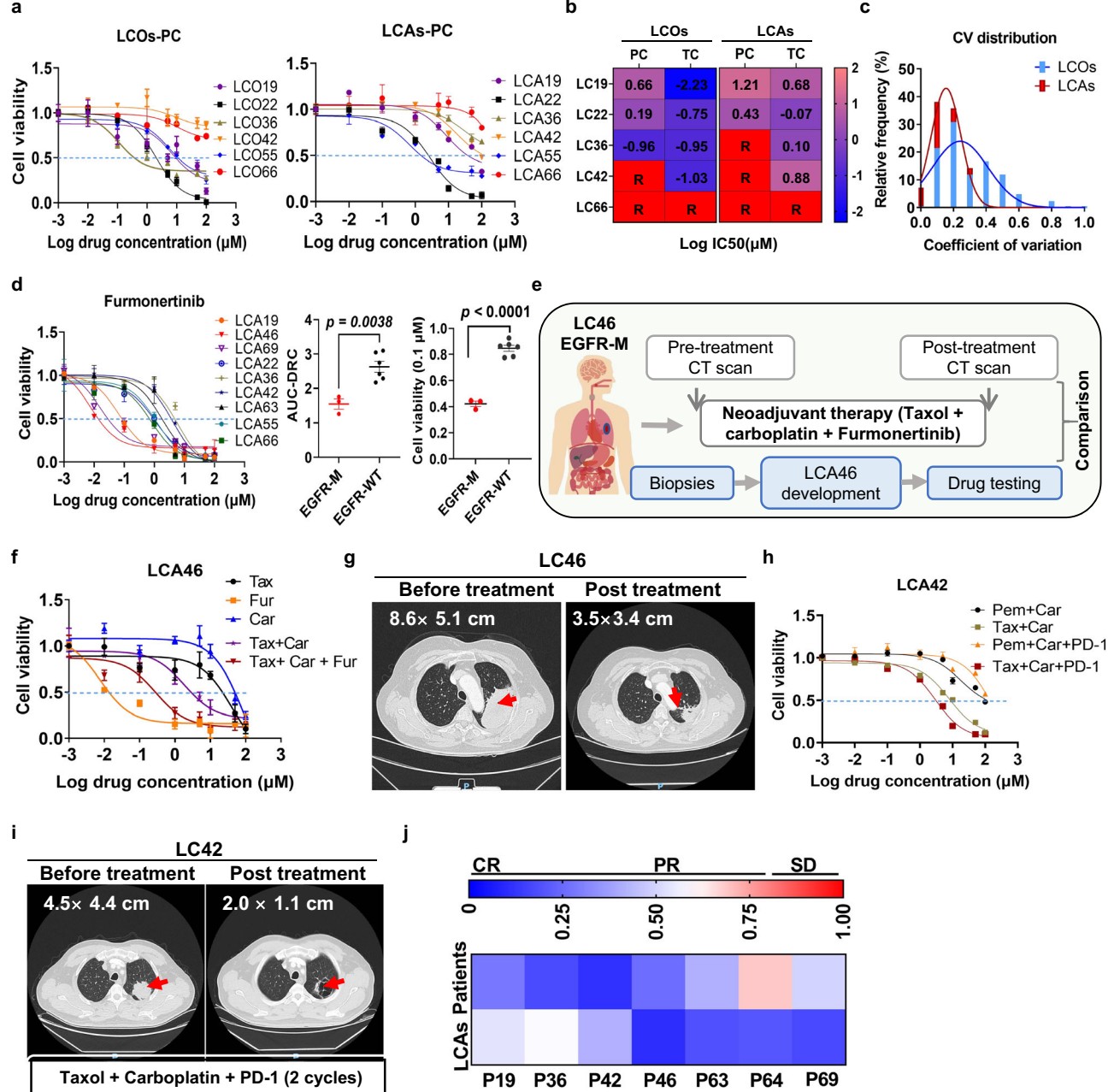

**Fig. 7 | LCAs as a powerful preclinical model for personalized drug testing.**
**a** Heterogeneous responses of LCAs and LCOs derived from 6 patients with lung adenocarcinomas to the carboplatin-based chemotherapies. The fitted dose−response curves illustrate the responses of the LCAs and LCOs to pemetrexed + carboplatin (PC) (*n* = 3 biologically independent samples). **b** The heat map is LogIC50 values of exemplary LCAs and LCOs in Fig. 7a and Supplementary Fig. 9a. TC, taxol + carboplatin. R means the IC50 is not available, and the samples are resistant to the drugs. **c** The distribution of CVs in the drug-sample pairs in the LCA and LCO groups. (*n* = 42 drug-sample pairs). **d** Dose response curves of various LCAs of nine different patients to Furmonertinib (*n* = 3 biologically independent samples). The right were the comparisons of both the area under the dose−response curve (AUC-DRC) and the viability at the indicated concentration (0.1 μM) between EGFR mutation (EGFR-M, *n* = 3 patients, LC19, LC46 and LC69) and the EGFR wild-type (EGFR-W, *n* = 6 patients) groups. **e** Flowchart of procedures

and treatments for LC46 patient with EGFR mutation. **f** Dose response curves of different drugs on LCA 46. **g** CT scan images of patient LC46 before and after neoadjuvant therapy. The size of the lung tumor, indicated by red arrow, decreased from 8.6 × 5.1 cm² to 3.5 × 3.4 cm² after neoadjuvant therapy (Taxol + Carboplatin + Furmonertinib). **h** The drug-response profile of LCAs from patient LC42. NC, negative control. Cis, Cisplatin. Pem, Pemetrexed (*n* = 3 biologically independent samples). **i** CT images of patient LC42 before and after neoadjuvant therapy. The size of the lung tumor, indicated by red arrow, decreased from 4.5 × 4.4 cm² to 2.0 × 1.1 cm² after 2 cycles of neoadjuvant therapy (Taxol + Carboplatin + PD-1). **j** The heat map of in vitro drug responses based on LCA test and clinical outcomes of 7 patients based on RECIST. CR, Complete response. PR, Partial response, SD, Stable disease. Scale bar, yellow bar, 200 μm; two-sided Student's t test is used and data are presented as mean ± S.E.M. Source data are provided as a Source Data file.

consistent drug responses across parallel wells in the LCA group compared to the LCO group (Supplementary Fig. 9c). Cell viability consistency across the parallel wells was further characterized systematically with the coefficient of variation (CV), and the LCA-based

drug testing results showed CV values less than 0.25 in 86% of cases vs. 59.5% in the LCO group, indicating more consistent results in drug-sample pairs of the LCA group (Fig. 7c). Additionally, correlation comparisons between any two sets of replications of drug screening

data in both LCA and LCO groups were performed. LCA group had mean Pearson's correlation coefficient 0.89 vs.0.74 and a mean $R^2$ (coefficient of determination) value 0.93 vs. 0.82 in LCO group (Supplementary Fig. 9d, e), demonstrating more consistent cell viability across replicated wells in LCA group. LCAs cultured within 2 weeks also showed good reproducibility of drug-responses between different independent experiments to both targeted therapy drugs and chemotherapy drugs (Supplementary Fig. 9f). These results above indicated that LCA model showed good reproducibility across parallel experiments and independent repeat experiments.

Next, we explored whether the LCA-based drug sensitivity tests could predict the patient responses to anticancer therapies. We firstly examined the effects of commonly used EGFR-targeted drug (Furmonertinib) on LCAs derived from 9 patient samples, three of which harbored EGFR activating mutations (EGFR-M), the other six had the wild-type EGFR(EGFR-W) (Supplementary Table 3). The drug–response curves of these 9 specimens were divided into two groups based on the sensitivities to Furmonertinib, perfectly accordant to their genetic mutations (Fig. 7d). We quantified the responses by calculating the area under the dose–response curve (AUC) and the relative viability at the indicated drug concentration (0.1 μM), both of which were significantly different between the EGFR-M and the EGFR-W groups ($P = 0.0038$ and $P < 0.0001$, respectively) (Fig. 7d). The interpolation of the drug–response curves gave half-maximal inhibitory concentration (IC50) values < 0.07 μM in the EGFR-M group, whereas >1 μM in EGFR-WT group (Supplementary Fig. 10a).

Next, we verified whether the LCA-based drug sensitivity tests can recapitulate the patients' responses to targeted therapy. LCA46 was generated from the biopsy of a lung cancer patient who carried the EGFR mutation before neoadjuvant therapy. We performed drug screening according to the clinical dosing regimen (Taxol + Carboplatin + Furmonertinib). The drug sensitivity tests indicated that LCA46 were more sensitive to Taxol + Carboplatin + Furmonertinib than to single-agent and double-agent chemotherapy, in agreement with the clinic outcomes (Fig. 7e–g). Additionally, LCA19 and LCA69 derived from patients carrying EGFR active mutation also showed consistent drug responses with their corresponding patients to the EGFR targeted drugs (Supplementary Fig. 10b–f). We also compared the responses to another ALK targeted drug (Ensartinib), between LCA63, which harbored the ALK positive expression, and the other two ALK-negative LCAs (LCA55 and LCA66). LCA63 was more sensitive to Ensartinib than the other two LCAs, indicated by the lower drug sensitivity curve and the >10 times smaller IC50 value (Supplementary Fig. 10g). These results above suggested that the LCAs could predict the patient responses to anticancer targeted therapies.

To assess whether LCA-based drug testing can predict patient responses to combined immunotherapy commonly used in clinic, LCAs of different patients were exposed to immune checkpoint blockade PD-1 plus platinum-based doublet chemotherapy. PD-1 plus platinum-based doublet chemotherapy could significantly improve the responses of LCAs derived from different patients (LCA36, LCA66 and LCA69) compared with platinum-based doublet chemotherapy group and PD-1 group (Supplementary Fig. 10h, l). This was consistent with the previous findings demonstrated that neoadjuvant PD-1 plus chemotherapy improved the event-free survival versus chemotherapy alone in patients with resectable non-small-cell lung cancer[62,63]. It's worth noting that consistent response results were observed in both LCAs-based drug assays and clinical outcomes of 3 patients with PD-1 plus chemotherapy. For example, both LCA42 and LCA36 were more sensitive to PD-1 plus TC treatment than to TC, consistent with the significant tumor shrinkage of the corresponding patients with 2 cycles of PD-1 plus TC treatment (Fig. 7h, i, Supplementary Fig. 10i). Of note, LC64 patient who received a combination treatment of PD-1 plus PC had a response evaluation of SD (stable disease) according to the images of positron emission tomography-computed tomography

(PET-CT) images, which was inconsistent with the LCA predictions (Supplementary Fig. 10j). However, histological examination supported that this patient had a significant pathological response with very few EpCAM+ tumor cells within the surgical tumor tissue (Supplementary. Fig. 10k), suggesting that judgments based on PET-CT were sometimes inaccurate[64].

Collectively, these results above indicated that LCAs could accurately predict the clinical outcomes of patients with an overall accuracy of 100% (7 of 7). (Fig. 7j). LCAs could be a promising preclinical model for personalized drug testing.

## Discussion

In vitro cancer models representing individual patients will facilitate the development of precision medicine and are urgently needed[65]. While patient-derived tumor organoids represent a new generation of in vitro tumor models that can be employed to predict the clinical outcomes of anticancer drugs, the lack of personalized TME and consistency limits their broad application[66]. Cancer assembloids three-dimensionally reconstituting cancer cells together with various cell types in the TME are thought to be more powerful models[21], but they suffer from the drawback of no efficient preparation methods, and few studies make a systematic characterization and quantitative comparison to the clinical response. Here, we report a method for the successful and high-throughput generation of LCAs that recapitulate the molecular, heterogeneous histology and TME features. We systematically compared the consistency of LCA-based drug testing results with the clinical outcomes of patients and assessed the value of LCAs as a preclinical model for personalized drug testing.

The TME is a critical component in tumors that significantly influences the therapeutic response and clinical outcome[67]. 3D cancer models with TME are critical for the understanding of cancer development and the development of anticancer drugs. Although some 3D approaches to model the TME, such as self-assembling spheroids, organoid coculture and organotypic models, have been developed[68], the lack of patient-specific TME, time consumption and poor intra-batch consistency limit their broad applications. In this study, we generated stable LCOs and corresponding TME cells from patient tumors and then precisely fabricated LCAs by evenly encapsulating TME cells and LCOs inside GelMA-Matrigel microgels with a droplet microfluidics-based method. The LCAs had patient-specific TME and intra-batch consistency, overcoming the limitations of traditional coculture and 3D bioprinting methods of assembloids[25]. Compared to lung cancer organoids, LCAs could recapitulate the patients' TME, including heterogeneous CAFs and the tumor immune microenvironment, facilitating the formation of tumor-like morphology and the applications in drug screening for combination therapy (Supplementary Table 1).

Due to the low and unpredictable efficiency of organoid derivation and expansion in vitro[69]. We always face the challenge of micro-volume samples (less than 100 μL) with limited cells or organoids. To overcome this challenge, we developed a microinjection strategy that enables us to manipulate microsamples precisely and fabricate LCAs with good cell distribution. Even 10 μL hydrogels containing $10^8$ cells mL$^{-1}$ could be successfully manipulated to generate ~200 uniform LCAs within 1 min. Two or three LCAs per well (~5000 cells) could be well tested for cell viability, indicating the potential of LCAs derived from tiny tumor tissues (e.g., biopsies) for personalized drug testing.

In addition, the good storability and sustainability of cancer models facilitate their broad applications in basic and translational research. We selected a GelMA-Matrigel composite hydrogel that ensures good biocompatibility, suitable mechanical features and good forming ability. More importantly, the LCAs formed with this hydrogel could be cryopreserved and thawed without little damage to the cells and LCA shapes, suggesting that LCA models could be bio-banked for future research. LCAs could also be digested into single cells with

gentle enzymes within 3 min for any other analysis. Nevertheless, we must admit the shortcomings of Matrigel-dependent hydrogels for cell or organoid culture. Matrigel is complex and poorly defined. Its complexity and lot-to-lot variation may bias cell selection and phenotype[9,70]. In addition, Matrigel may exhibit gradual diffusion out of LCAs over culture time, leading to a potential decrease in consistency after a relatively long-term culture. We will develop some alternative Matrigel-free materials for the generation of LCAs in the future.

Good consistency between drug testing and patient responses in the clinic is the gold standard for a preclinical model. Although some cancer organoids[71,72] and patient tumor-derived cell clusters[11] were reported to recapitulate patient responses in the clinic, the lack of inter- and intra-batch consistency and heterogeneous TME affects the efficiency and accuracy of drug testing[73]. Our LCA model demonstrated higher consistency across the wells of parallel experiments than LCO model, while the reproducibility of drug response phenotypes for both LCAs and LCOs seem to be high. Additionally, LCAs and LCOs showed different responses to chemotherapeutic and targeted drugs for the presence of tumor micro-environments. And these comparisons between LCOs and LCAs need be further performed in larger numbers of samples in the future. In addition, our LCA model could accurately predict clinical treatment outcomes of patients with neoadjuvant immunotherapy in combination with chemotherapy and targeted therapy. Even though, we have to acknowledge that the current sample size is not sufficient. We will further validate the accuracy of LCA model in predicting clinical drug responses especially for immune checkpoint blockade therapy in more samples in the future.

In summary, we provide here a promising method for the high-throughput generation of uniform cancer assembloids and a personalized preclinical model that replicates patient-specific TME as well as other key features of parental tumors. This model has the potential for testing personalized treatment responses and broad applications in basic and translational research. This model and future adaptations may drive clinical-translational efforts to develop combination therapies and personalized therapy for lung cancer.

# Methods

## Human tumor processing
All lung cancer samples were collected from Peking University Cancer Hospital and Chinese Academy of Medical Sciences & Peking Union Medical College under a protocol approved by the Medical Institutional Review Board of Tsinghua University (accession number: 20220301). All patients gave written informed consent. Lung tumors and their adjacent normal lung tissues were stored in MACS tissue storage solution (Miltenyi Biotec) and transported to the laboratory within 2 h. The tissues were cut into three parts (1 mm$^3$). One part was rapidly stored in liquid nitrogen for subsequent whole-genome DNA sequencing and RNA sequencing. One part was fixed in 4% paraformaldehyde for histopathological and immunohistochemical staining. The rest was digested for organoid and TME cell generation. The needle biopsies of tumors were digested directly for organoid culture or LCA fabrication.

## Lung cancer organoid (LCO) culture
To generate LCOs from the tumor tissue of lung cancer patients, we used DPBS containing 1 × penicillin–streptomycin solution to wash the tumor tissue three times. We minced the tumor tissue into small fragments of 1 mm$^3$ in a 6 cm cell culture dish using surgical scissors. Then, the small tumor pieces were digested using 5 mL collagenase (2 mg ml$^{-1}$ each of collagenase I and IV) on a shaker for 1 h, and the contents were dispersed every 15 min by pipetting the mixture up and down using a P1000 pipette. The digestion was terminated by Advanced DMEM/F12 (Thermo Fisher Scientific) supplemented with

10% FBS (BI). The digested cell solution was filtered using 300 μm and 150 μm strainers. The filtered cells or cell clusters were washed with PBS and centrifuged. We then quickly aspirated the supernatant and resuspended the pellet in Matrigel (Corning, pellet: Matrigel ≈ 1:3). Matrigel was kept on ice to avoid solidification. Matrigel-containing cells were plated on the bottom of 24-well plates (preheated at 37 °C) in droplets of 10 - 20 μL each and incubated on the culture plate at 37 °C in 5% CO2 for 10 min. Once the drops were solidified, 500 μL of LCO medium was added to the wells, and the plate was transferred to a cell culture incubator at 37 °C with 5% CO2. The LCO medium was refreshed every 3 days. The recipe for the LCO medium is listed in Supplementary Table 5.

## TME cell isolation and culture
CAFs could be isolated from the organoid culture system via the differential adherent method when LCOs were passaged for the first time. Passaging of LCOs was performed using the mechanical method. Briefly, LCOs were harvested and suspended in cold PBS at 4 °C for 20 min to dissolve the Matrigel. Meanwhile, the attached CAFs were digested with Tryple (Thermo Fisher Scientific) for 3 - 5 min, followed by washing and culturing with DMEM (Thermo Fisher Scientific) containing 10% FBS.

Once the Matrigel was dissolved completely, the suspension was pipetted several times to thoroughly disperse the single cells and LCOs. Then, the suspension was processed for differential sedimentation for 2 min, and the upper layer of the cell suspension containing TILs was transferred to a new tube, followed by washing, centrifugation (400 × g, 5 min, 4 °C), and culture in RPMI 1640 medium (Thermo Fisher Scientific) supplemented with 5% FBS, 600 IU mL$^{-1}$ IL-2 (Novoprotein), 100 U mL$^{-1}$ penicillin, and 100 mg mL$^{-1}$ streptomycin at 37 °C and 5% CO2. The medium was replaced every 3 days, and TILs were passaged at 1:2 on Day 6 or when necessary. LCOs that settled to the bottom of the tube were digested with Tryple for <15 min and then reseeded at a ratio of 1:2–1:4 in Matrigel.

## Cell culture
A549 cells were purchased from the American Type Culture Collection (ATCC, USA, CRM-CCL-185). Cells were cultured in RPMI 1640 medium with 10% FBS, 1% penicillin and 1% streptomycin (Invitrogen). Cell passage was done every 3 days. The cells were cultured at 37 °C in a humidified incubator with 5% CO2.

## Microfluidic platform setup
The microfluidic platform was set up by adapting our previous work[74]. Briefly, the microfluidic platform mainly consisted of two infusion pumps, a T-junction PDMS chip, a lighting module, a microscope module, and a heating module. The T-junction PDMS chip was fabricated by soft lithography, connected with a piece of silicone tubing (Woer, ID = 0.56 mm) with a blunt end G22 needle. Both the hydrogel precursor and oil phases were loaded into the silicone tubes, which were separated by air bubbles and further pushed by the aqueous solution in the 1 mL plastic injection syringe controlled by the infusion pumps. Microchannels of the T-junction PDMS chip were first primed with mineral oil (Sigma, Germany) supplemented with 2% span 80 at the flow rate of 5 mL h$^{-1}$. The cell-laden polymer precursor solution was heated to be around 37 °C by a heating module, injected and sheared into monodisperse droplets at the merging points of both fluids. The polymer droplets were in situ crosslinked using the lighting module by exposure to visible light at 405 nm at the outlet. The microgel fabrication process was monitored in real-time by a handheld microscopy, which was connected to the computer by USB. It should be noted that the microfluidic platform is assembled within a clean bench and sterilized using ultraviolet light. The flow rates of both phases were controlled by the infusion pumps to adjust the sizes of cell-laden microgels.

## Fabrication of lung cancer assembloids

Tumor organoids and TME cells obtained from human tumors were mixed into the GelMA-Matrigel hydrogel. Then, we added PBS to a 1 mL syringe attached to a hollow silicone injection tube and installed the syringe in a water-phase syringe pump. Mineral oil (Sigma) with 2% span80 (Sigma) was loaded into a 20 mL syringe and installed in an oil-phase syringe pump. The cell-laden hydrogel was sucked into the silicone tube by a water-phase syringe pump. The hydrogel and PBS were separated by a little air. The end of the silicone tube was then connected to the T-junction polydimethylsiloxane (PDMS) chip, where cell-laden hydrogel was sheared by mineral oil into monodisperse droplets. The flow rate of the water phase was $1 \, \text{mL h}^{-1}$ and the oil phase was $5 \, \text{mL h}^{-1}$. Then, the droplets were cross-linked by the 405 nm UV cross-linking module to form LCA precursors. The precursors were collected in a 15 mL centrifuge tube and resuspended in DMEM containing 10% FBS, followed by centrifugation at $250 \times g$ for 3 min. We discarded the oil phase and rinsed the LCA precursors in DMEM again and then resuspended the precursors using LCA culture medium containing LCO medium, CAF medium and TME medium at a ratio of 2:1:1. Finally, the LCA precursors were transferred to an ultralow attachment 24-well cell culture plate and then cultured for 3 days to form LCAs in a 37 °C and 5% CO2 incubator.

## GelMA-matrigel hydrogel preparation

GelMA (EFL, China) was dissolved in PBS containing 1% LAP in the dark for 30 min in a 37 °C water incubator. GelMA-Matrigel hydrogels were prepared by adding GelMA hydrogel to Matrigel diluted with PBS. The mixture of hydrogels containing different concentrations of Matrigel (15% and 30%, v/v) was incubated at 37 °C before use. For the cell-laden hydrogels, the centrifuged cells were suspended in the GelMA-Matrigel hydrogel and mixed well. The cell-laden hydrogel was incubated at 37 °C before being loaded into the droplet microfluidics.

## Mechanical testing

The compressive mechanical properties of GelMA-Matrigel hydrogels and tumor tissues were measured by a mechanical testing machine (Bose ElectroForce 3200, Bose Corp.). Photo-crosslinked hydrogels with a length and width of 7.4 mm and height of 2 mm were fabricated with the perfusion model under the same conditions. Before all measurements, all samples were measured with a Vernier caliper to determine their actual sizes. Stress–strain curves were obtained by normalizing the cross-sectional area and height of the samples from the loading and displacement data, and the elastic modulus was calculated from the linear region of the resulting stress–strain curves (10 - 20% of strain).

## Cell viability

Cell viability was assessed using the Calcein-AM/PI double-staining kit (Dojindo). In brief, cells were incubated with 2 μM calcein-AM, along with 3 μM PI in DPBS for 20 min at 37 °C, followed by washing twice with DPBS after incubation. Fluorescence images were taken with a confocal microscope (FV3000, Olympus). Live and dead cells were observed as green and red fluorescent signals, respectively. According to the microscopic scanned picture, the cell viability was quantified using Fiji ImageJ software.

## Cell proliferation assay

The cell proliferation ability was evaluated using the Cell Counting Kit-8 (Dojindo) by measuring the metabolic activity of surviving cells, according to the manufacturer's instructions. We diluted the CCK-8 agent at 1:10 in fresh medium and used it to treat the assembloids. After 4 h of incubation, 100 μL of medium was transferred to a 96-well plate. The absorbance was measured at 450 nm using a microplate reader (Multiskan FC Microplate photometer, Thermo Scientific). The optical density (OD value) for absorbance was directly proportional to

the number of living cells. For each condition, at least three samples were tested.

## Immunofluorescence staining

LCAs and tumor fragments were fixed with 4% paraformaldehyde (PFA, Solarbio, China) for 30 min at room temperature. All samples were blocked and permeabilized using 10% (w/v) goat serum solution (Solarbio, China) with 0.2% Triton X-100 (Sigma) for 1 h at room temperature on a shaker. Samples were then incubated with the respective primary antibodies overnight at 4 °C. On the next day, samples were rinsed with PBS with 0.05% Tween 20 (PBST) three times (10 min each time) on a shaker followed by incubation with fluorophore-conjugated secondary antibodies (1:500) at room temperature in the dark. Finally, the samples were counterstained with DAPI and visualized using a laser scanning confocal microscope (FV3000, Olympus). The data were collected and analyzed in FV31S-SW Viewer software (Olympus). Information on the antibodies is listed in Supplementary Table 7.

## Detection of hypoxia gradients in LCAs

LCAs were incubated in LCA medium containing 200 mM pimonidazole-HCl (Hypoxyprobe, USA) for 3 h on a shaker rotating at 60 rpm in a 37 °C, 5% CO₂ incubator and processed for normal immunofluorescence staining. Mouse anti-pimonidazole monoclonal antibody conjugated to FITC was used to detect bound pimonidazole.

## Apoptosis analysis

Cell apoptosis analysis in LCA was performed by using an Annexin V-FITC Apoptosis Detection Kit (YEASEN, China) according to the manufacturer's instructions. In brief, LCAs were dissociated with trypsin containing 0.25% EDTA (Sigma) and washed in PBS buffer. A total of $2 \times 10^5$ cells were collected and resuspended in Annexin V binding buffer (200 μL), and Annexin V-FITC (5 μL), PI (5 μL) and anti-human EpCAM-APC (5 μL, marking the tumor cells in LCAs) were added. The cells were incubated at room temperature for 15 min in the dark and then resuspended in binding buffer (300 μL) after centrifugation. Finally, cell apoptosis was analyzed via flow cytometry (BD LSRFortessa, BD Biosciences) and the data was analyzed in FlowJo (v10).

## RNA-seq and Data Preprocessing

An RNA library was constructed and sequenced using the Illumina NovaSeq6000 platform with paired-end reads according to the manufacturer's instructions. The data quality was assessed by FastQC (v0.11.5). Low-quality reads and adapters were removed by Trimmomatic (v0.39)[75]. Then, hisat2 (v2.1.0) was used to align reads to the reference genome GRCh38. Finally, HTseq (v0.13.5)[76] was utilized to yield read counts. The R packages limma[77] and edgeR[78] were used to generate the TMM[79] normalized log(CPM) (log2 counts per million) expression of the genes. Sample similarities were measured by the Pearson correlation coefficients (R) of sample log(CPM) which were calculated using the cor function in R. Then, the differentially expressed genes were selected (upregulated: log fold change > 1.5, down-regulated: log fold change < −1.5). The differentially expressed genes shown in the heatmap (Fig. 4a) were selected as follows: for each patient, we identified the differentially expressed genes as the commonly upregulated or down-regulated genes in tumor and LCA compared to normal by log fold change (>1.5 or <−1.5); and then, the differentially genes for all the four patients (LC14, LC28, LC51 & LC52) were combined in the Fig. (6577 genes in total). Gene set enrichment analysis (GSEA) was performed by Metascape[80].

## Whole exome sequencing (WES) analysis

Tumors, paracancerous tissues, and corresponding LCAs and LCOs were snap-frozen on dry ice and stored at 80 °C until processing. DNA was extracted using the M5 AllPure DNA/RNA/Protein Kit (Mei5bio,

China). Exome sequencing was performed by Geekgene (Beijing, China) using the Illumina NovaSeq 6000. The original data were trimmed into splice sequences and then filtered using Trim Galore (v0.6.7) and aligned to the human reference genome GRCh38 using BWA (v0.7.17). GATK (v4.2.2.0) was used to label PCR duplicates and base quality recalibration according to GATK best practices. Somatic variants were identified using Mutect2 (from GATK v.4.2.2.0) and further annotated by ANNOVAR (v2020.06.08)[81]. The variant annotations from ANNOVAR were then converted into MAF files, and the visualization of top mutated genes and variant classification was performed by the R package Maftools[82]. Copy number alterations were evaluated with CNVkit (v0.9.10)[83].

### Sample preparation and single-cell RNA sequencing

Tumor fragments and LCAs were dissociated using TrypLE™ (Gibco). Crude dissociates from parental tumor samples were treated with RBC lysis buffer (Thermo Fisher Scientific) for 5 min. All samples were washed three times by centrifugation at $200 \times g$ for 5 min and resuspended in 10 mL of calcium-free, magnesium-free PBS (Gibco). Cells were strained through a 40 μm filter (Corning), analyzed for viability by trypan blue staining, and counted using an automatic cell counter (Thermo Fisher Scientific). Samples had a viability of > 85% and were diluted to a final concentration of 100 viable cells/mL in PBS with 0.01% BSA (w/v). The single-cell suspension was loaded into Chromium microfluidic chips with 3' (v2 or v3, depending on project) chemistry and barcoded with a 10 × Chromium Controller (10 X Genomics). RNA from the barcoded cells was subsequently reverse-transcribed, and sequencing libraries were constructed with reagents from a Chromium Single Cell 3' v2 (v2 or v3, depending on project) reagent kit (10X Genomics) according to the manufacturer's instructions. Sequencing was performed with Illumina (HiSeq 2000 or NovaSeq, depending on project) according to the manufacturer's instructions (Illumina).

### scRNA-seq data processing

The raw scRNA-seq data (FASTQ files) were aligned to the hg38 reference genome using Cell Ranger (version 7.0.0). The output filtered_feature_bc_matrix folders were directly processed by the R package scCancer (v2.2.1)[84] for basic QC, downstream analysis, and data integration. Specifically, quality control was implemented on the expression matrices, and suggested thresholds for filtering cells and genes were obtained by running the scStatistics function. Then, the scAnnotation function was run to implement Seurat pipelines (log-normalization, finding highly variable genes, scaling, PCA, SNN graph construction, and clustering) and perform some cancer-specific analyses, including doublet score estimation, tumor microenvironment cell type classification, and hallmark gene set signature analysis. Finally, six datasets were merged by the scCombination function using harmony (v1.0)[85] as the batch correction method with the default settings.

To comprehensively characterize the cell composition in tumors and LCAs, we followed the Seurat (v3.9.9)[86] pipeline and conducted two rounds of clustering based on harmony reduction. In the first round, major cell populations were found based on canonical markers (*EPCAM, COL1A1, CD3D, CD79A, CPA3, LYZ, and CD68* for epithelial cells, fibroblasts, T cells, B cells, mast cells, and macrophages). Clusters with low counts (<600) and feature numbers (<500) as well as no expression of major cell markers were labeled empty droplet clusters and excluded from the downstream analysis. In the second run, cells from three major cell types (epithelial cells, fibroblasts and T cells) were reclustered. Due to the patient-specific nature of epithelial cells, epithelial cells from two donors were reclustered and annotated separately. The cluster with a high median doublet score (>0.8) and expression of two major cell markers (*EPCAM and COL1A1*) was considered a doublet cluster and removed. Differentially expressed genes (DEGs) in each clustering run were detected by Seurat with the default settings. Finally, we obtained 51479 fine-annotated cells according to the DEGs and cell markers of subpopulations (see dotplot in Supplementary Fig. 6b).

Analysis and visualization of cell interactions were performed using the R package CellChat (1.4.0), which calculated communication probabilities using the law of mass function and inferred the biologically significant cell-cell communication using permutation tests. The cell type labels used were derived from the fine clustering result. We followed the tutorials (https://github.com/sqjin/CellChat) and used default parameter settings[87].

### Enzyme-linked immunosorbent assay (ELISA)

The levels of HGF and FGF7 secreted by CAFs were measured via ELISA. After 48 h of culture, culture supernatants of CAFs were collected and then immediately frozen and preserved at −80 °C. Supernatants were thawed on ice, and the presence of HGF and FGF7 was detected using the HGF (Solarbio, China) and FGF7 Human ELISA Kit (PYRAM, China). HGF and FGF7 were further normalized to IGFBP-6, a stably expressed cytokine with levels directly proportional to the raw count of CAF cells. The ELISA data were collected on a SpectraMax M2 Microplate Readers (Molecular Devices).

### Drug treatment assay

Drug testing was conducted in ultralow attachment 96-well cell culture plates (Corning). Briefly, the LCAs or LCOs encapsulated in GelMA-Matrigel cultured for 3 days were changed with fresh media. Then, medium containing 10 ~ 20 LCAs (80 μL) was seeded into a 96-well ULA plate (Corning) using a multichannel pipette (Eppendorf). Next, LCA medium containing the drugs (30 μL) was added to the wells. The plates were incubated at 37 °C for 3 days, and the cell viability was measured by using CellTiter-Glo 3D reagent (Promega) according to the manufacturer's instructions. The readout was performed by measuring the luminescence signal by using SpectraMax M2 Microplate Readers (Molecular Devices). The chemotherapy and targeted therapy drugs are listed in Supplementary Table 4.

### Statistical analysis

Data were analyzed by one-way analysis of variance using GraphPad Prism software (GraphPad, La Jolla, CA) and are described as the mean ± S.E.M. Statistical analysis was performed by two-tailed unpaired $t$ tests for comparisons of two groups. Significant differences between groups were noted as follows: $*p < 0.05$, $**p < 0.01$, and n.s, $p > 0.05$, not significant.

### Reporting summary

Further information on research design is available in the Nature Portfolio Reporting Summary linked to this article.

## Data availability

All data are available in the article and its Supplementary files. Source data are provided with this paper. All the raw data for the RNA-seq, exome-seq and single-cell RNA-seq data reported in this study have been deposited in GSA-Human database (Genome Sequence Archive for Human, as a part of GSA in the National Genomics Data Center) and are publicly available for all readers by visiting GSA-Human (https://ngdc.cncb.ac.cn/gsa-human/) (accession numbers: HRA003470 and HRA004052) under the approval of the Ministry of Science and Technology (accession number:2023BAT0242 and 2023BAT1141). Source data are provided with this paper.

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

## Acknowledgements

The authors thank all patients who generously donated their tissues for this study. The authors thank Yue Sun and Jingjing Wang at Cell Biology Facility, Center of Biomedical Analysis, Tsinghua University for the help with confocal microscopy. This work was Sponsored by Tsinghua-Toyota Joint Research Fund (No. 20223930093, Z.X.) and grants from the National Nature Science Foundation of China (No. 52305298, Y.Z.).

## Author contributions

Z.X. and Y.Z. conceived and supervised the study. Q.H. contributed to bioinformatics data analysis and data integration of all the sequencing data. Y.P. contributed to sample collection and collection of patient clinical information. H.L. contributed to the development of micro-injection strategy-based droplet microfluidic technology. Y.Z., H.L., Z.W., X.X., Q.Z., J.D. contributed to data processing and generating figures and tables. Q.W. and Z.F. contributed to a part of LCA fabrication and culture. M.Y. and Y.F. assisted with data processing and manuscript revision. B.L., M.C., Q.X., Q.Z., M.H. and S.Z. assisted with clinical sample collection. T.Z. contributed to the guidance for some experiments. J.G. provided the guidance of bioinformatics experiments and bioinformatics data analysis. Y.Z. wrote the manuscript. Z.X., T.Z., H.L., Z.W., Q.H., Q.W., Y.M. and Y.F. revised the manuscript.

## Competing interests

The authors declare no competing interests.

## Additional information

[1]Biomanufacturing Center, Department of Mechanical Engineering, Tsinghua University, Beijing 100084, China. [2]Biomanufacturing and Rapid Forming Technology Key Laboratory of Beijing, Beijing 100084, China. [3]Biomanufacturing and Engineering Living Systems Innovation International Talents Base (111 Base), Beijing 100084, China. [4]Institute of New Materials and Advanced Manufacturing, Beijing Academy of Science and Technology, Beijing 100089, China. [5]MOE Key Laboratory of Bioinformatics, BNRIST Bioinformatics Division, Department of Automation, Tsinghua University, Beijing 100084, China. [6]Key Laboratory of Carcinogenesis and Translational Research (Ministry of Education), Department of Thoracic Surgery II, Peking University Cancer Hospital and Institute, Beijing 100142, China. [7]Medical School of Chinese PLA, Beijing 100853, China. [8]Department of Radiology, Peking University Cancer Hospital & Institute, Beijing 100142, China. [9]Department of Thoracic Surgery, National Cancer Center/National Clinical Research Center for Cancer/Cancer Hospital, Chinese Academy of Medical Sciences and Peking Union Medical College, Beijing 100021, China. [10]These authors contributed equally: Yanmei Zhang, Qifan Hu, Yuquan Pei. ✉e-mail: jgu@tsinghua.edu.cn; xiongzhuo@tsinghua.edu.cn

