## [Peer Review File · Nature Communications]

A patient-specific lung cancer assembloid model with heterogeneous tumor microenvironmentsREVIEWER COMMENTS

Reviewer #1 (Expertise: Microfluidic technology and lung cancer organoids, Remarks to the Author):

The authors describe generation of patient-specific lung cancer assembloids (LCAs) generated by microinjection-based droplet microfluidic devices. These assembloids are generated by flowing patient-derived tumor organoids generated in Matrigel in a Matrigel/gelatin-methacrylate solution into a mineral oil containing channel in a microfluidic device, thus generating droplets of the hydrogel precursor, which are then photopolymerized through the methacrylate groups on the gelatin components. The LCAs are characterized in a number of ways, including biomarker expression, viability, hydrogel mechanical properties, consistency, transcriptomics, and drug response. There are some nice comparisons with patient responses to treatments that were correspondingly used in the LCAs, showing correlation between treatments. However, this is only done in 3 or 4 cases.

The manuscript is well organized, as are most of the experiments. There are some high points – such as the transcriptomics, great imaging, and the patient-to-LCA comparisons. However, these data sets are only described for several LCA sets. Moreover, from the 33 biospecimens collected, only several are presented, and it is not clear how many LCA sets were successfully generated. Other concerns include the field-wide reliance on Matrigel, which can bias experiments due to its undefined nature, the potentially overcomplication of needing the microinjection microfluidic method to form the LCAs, and generalized statements about limitations of bioprinting and other organoid/tumor model limitations that are not quite accurate.

Specific Comments:

1. There are some grammatical errors throughout. Please proofread the manuscript more thoroughly.
2. Introduction, line 72-78. The statement that organoids only represent tumor epithelium is not accurate. There are many examples of traditional Matrigel-based tumor organoids with heterogeneity, as well as other non-traditional organoid/tumor constructs that maintain heterogeneity. This should be acknowledged.
3. Introduction, line 86-89. It is not very challenging to use extrusion bioprinting to create small volume tumor constructs. 10 uL or even 5 uL is quite straightforward with most of today's 3D bioprinters. Moreover, this approach can be use in a medium-to-high-throughput manner for rapid generation of tumor models. This is counter to the authors' statements.
4. Figure 1 is missing labels (e.g. (a), (b), etc.).
5. The microinjection method seems unnecessarily complicate. Since the assembloids are simply organoids and other cells mixed together that form essentially a heterogeneous spheroid, couldn't "assembloids" be formed by hanging drop culture or culture in ultralow adhesion round bottom wells? Or by 3D printing small hydrogel droplets and crosslinking those?
6. It is certainly true that Matrigel is a potent matrix for supporting tumor cells and generating patient-derived organoids. However, it also is undefined and can essentially introduce variables that unintentionally can bias experimental results. Additionally, it is derived from a murine sarcoma tumor, which is inherently different than lung tumors – and most other tumors. The authors should acknowledge these problems.
7. Mechanical analysis results: It should be stated in the results what kind of mechanical analysis was performed. Rheology? Indentation? Compression?
8. Results, lines 136-138. While technically correct, this is a bit of an overstatement. Using bioprinters that have been on the market for 5 years already, several hundred hydrogel-based

tumor constructs can be created in several minutes. The microfluidic method is not significantly faster.

9. Other in vitro 3D tumor models are also bankable, so this is not a unique feature.

10. Figure 3: In panel b, some of the LCAs look very acellular. Others look highly irregular. This goes against the "consistency" touted by the authors.

11. Supplementary 4: Panel b – It's not clear if this is a fair comparison. There appears to be a lot more cellular material in the U-plate image than in the individual droplets. Moreover, in panel c, the U-plate merged panel doesn't look all that different than the LCAs.

12. RNAseq data: Since the genetic shift occurs over time like in organoids, how are the LCAs better than organoids?

13. Figure 6. The patient comparison in panels e-g is nice. However, LCOs have TIL here, making them almost cellularly on par with the LCAs. The thought is that the TME adds resistance to the LCAs. Which part of the TME?

14. The data has LC's numbered up to LC42, yet only a selection are included in the presented data. How many samples were collected and how many generated successful LCOs/LCAs? Out of the successfully established LCOs/LCAs, how many recapitulated patient clinical results. Without these numbers it's impossible to assess whether this system was indeed successful, or if it was only predictive in 3 patients.

15. The gelatin-methacrylate is photopolymerized, but the Matrigel is in a flowable form. How do the authors know that the Matrigel components don't slowly diffuse out of the assembloids. Validation of this should be included.

Reviewer #2 (Expertise: cancer genomics, Remarks to the Author):

In the manuscript entitled "A patient-specific lung cancer assembloid model with heterogeneous tumor microenvironments", Zhang et al. proposed a droplet based microfluidic technology to generate lung cancer assembloids (LCA). With the ability to mimic the tumor microenvironment of in vivo tumors, such in vitro cancer models seems to be very promising for the development of personalized therapeutic approach. The authors validated how LCAs recapitulated actual tumors, in terms of the genomic/transcriptomic landscapes, TME composition etc. Overall, I do think the manuscript present an exciting advancement, I have a few comments in terms of improving the analysis and presentation.

Major:

1. I think it would be helpful if the authors could mention some basic parameters of the LCA (such as the mean diameter and the mean number cells involved) at the beginning of the text, even though numbers are already scattered around in figures and in the discussion.
2. Is it possible perform single-cell RNA-seq for cells in the LCAs? It seems to be very helpful to profile the cellular composition of LCAs. If not, what are the challenges? The authors should at least discuss that in the manuscript.
3. Even though single-cell transcriptomic profiling could be a challenge, it is still possible to shed light on cellular composition. The authors could find the expression profiles of CAFs in lung cancer as reference, and then perform deconvolution on their bulk RNA-seq experiments
4. The effects of TME on tumors are mostly mediated by ligand-receptor interactions. Using curated ligand-receptor pairs, the authors can examine if the interactions are preserved from patient tumor to LCA using their bulk RNA-seq data. It could provide better understanding on the roles of CAFs in the models, as compared to the patient tumors.
5. An alternative cancer model that could capture the effects of TME is patient-derived xenograph, can the authors discuss compare LCA and PDX in terms of their usage as cancer models?
6. One important feature of the LCA model claimed by the authors is that this model could

maintain the cell composition of parental tumors. However, the authors only checked a few cell type-specific markers in a few samples. Considering the preparation of bioink, it's expected to see the markers shown in Fig. 1. 1) What are the patterns in other samples? Are they similar with those shown in Fig.1? 2) What are the proportions of tumor cells, CAFs and TILs in parental tumors and LACs? Single-cell RNA-seq may help.

7. One of the most significant drawbacks of conventional cancer organoid models, as mentioned by the authors at Line 72-74, is that these models can only represent tumor epithelium, and they loss stromal and immune cells gradually over time in culture. However, the changes over time in culture were also observed in LCA model (e.g. Fig. 4b). It's of great significance to investigate more on these over-time changes of the LCA model: 1) how long could the main features (gene expression/cell composition/CAF heterogeneity/performance of drug test) be maintained? 2) how do these features change over time in culture? This should definitely be mentioned in the Discussion.

Minor:

1. Line 184-186: please provide the correlation coefficients to support the statement that the transcriptome profiles are similar between LAC and corresponding parental tumors. It's not obvious in Fig. 4a.

2. Line 190-193: the authors provided the top enriched pathways of the genes up-regulated in LCA and tumors compared to normal samples. While the more interesting and meaningful comparison would be between LCA and tumors. Please perform the pathway enrichment analysis with the up-regulated genes in LCA (v.s. normal) and parent tumor (v.s. normal), and see how similar they are.

3. Fig. 1: the numbering of panels, a, b and c, are missing.

4. Fig. 4a and e: the color of the bars indicating Normal, Tumor and LCA should be consistent between LC14 and LC28.

5. Fig. 4f: how these 63 genes were selected?

6. Fig. 4g: redundant with Fig. 4f. It might be better to just highlight these genes in Fig. 4f.

7. Fig. 4h: the heatmap needs to be properly scaled. Currently it's not intuitive to say the CNV profiles are similar between T and A. Can the authors comment on that?

8. Line 273, the conclusion is not too obvious as seen in Supplementary Fig. 7a.

9. Fig. S7b: it might be more appropriate to use grouped bars instead of stacked bars.

RESPONSE TO REVIEWERS' COMMENTS

Here we present a detailed point-by-point response to the comments raised by the reviewers.

Reviewer #1 (Remarks to the Author):

The authors describe generation of patient-specific lung cancer assembloids (LCAs) generated by microinjection-based droplet microfluidic devices. These assembloids are generated by flowing patient-derived tumor organoids generated in Matrigel in a Matrigel/gelatin-methacrylate solution into a mineral oil containing channel in a microfluidic device, thus generating droplets of the hydrogel precursor, which are then photopolymerized through the methacrylate groups on the gelatin components. The LCAs are characterized in a number of ways, including biomarker expression, viability, hydrogel mechanical properties, consistency, transcriptomics, and drug response. There are some nice comparisons with patient responses to treatments that were correspondingly used in the LCAs, showing correlation between treatments. However, this is only done in 3 or 4 cases.

The manuscript is well organized, as are most of the experiments. There are some high points – such as the transcriptomics, great imaging, and the patient-to-LCA comparisons. However, these data sets are only described for several LCA sets. Moreover, from the 33 biospecimens collected, only several are presented, and it is not clear how many LCA sets were successfully generated. Other concerns include the field-wide reliance on Matrigel, which can bias experiments due to its undefined nature, the potentially overcomplication of needing the microinjection microfluidic method to form the LCAs, and generalized statements about limitations of bioprinting and other organoid/tumor model limitations that are not quite accurate.

Response: We would like to thank the reviewer for the detailed and valuable comments on our manuscript that we used to improve our work.

Indeed, the comparisons of drug responses between 4 pairs of LCAs and corresponding patients, were insufficient to draw definitive conclusions. Therefore, we have done 4 more comparisons and now a total of 8 comparisons between patients and LCAs are included in this revised manuscript (Revised Fig. 7m; Suppl. Fig. 10).

Additionally, following the review's suggestion, we have provided a clear list that distinguishes the successful and unsuccessful LCOs/LCAs in revised Supplementary table 1. A total of 49 clinical tumor samples were collected and tumor-derived cells of 36 patients were used to fabricate LCAs by the droplet microfluidic method. We successfully fabricated LCAs of 35 patients with a 97.2% success rate. This is now stated explicitly in the revised manuscript (line 118-121).

We acknowledge and concur with the concerns raised by the reviewer regarding Matrigel, a hydrogel derived from the secretion of mouse sarcoma cells and enriched for extracellular matrix (ECM) proteins¹. Its undefined culture system, batch-to-batch variation and poor control of mechanical properties are the critical limitations of Matrigel². However, it's worth noting that Matrigel is still the prevailing standard culture system of organoids to date due to its biomimetic property and versatility³⁻⁶. In our manuscript, we acknowledged the drawbacks of Matrigel such as poor control of mechanical properties and undefined effects. In light of these considerations, we chose GelMA as the base hydrogel for its excellent processing capability, tunable mechanical properties, and biocompatibility even for immune cells⁷⁻¹¹. However, cells from primary tumors could not grow and proliferate well in GelMA hydrogel. The addition of Matrigel could significantly improve cell viability and proliferative capacity. We also found that low proportion of Matrigel (6-15 group, 15% Matrigel included) could ensure both the biomimetic mechanical property of hydrogel and good viabilities of primary cell (Fig. 2d-g; Supplementary Fig. 3a-e). Therefore, we used 6-15 GelMA-Matrigel hydrogel as the bioink to fabricate LCAs. Our manuscript focused on the precise construction of cancer assembloids using cells or organoids derived from patient tumors. In the future work, we will find or develop Matrigel-independent system to fabricate and culture cancer assembloids.

On the other hand, we have updated the statements about limitations of bioprinting and other organoid/tumor model limitations according to the latest literature in this revised manuscript.

Specific Comments:

1. There are some grammatical errors throughout. Please proofread the manuscript more thoroughly.

Response: Many thanks for your kind reminder. We have carefully checked possible language issues and made necessary corrections throughout.

2. Introduction, line 72-78. The statement that organoids only represent tumor epithelium is not accurate. There are many examples of traditional Matrigel-based tumor organoids with heterogeneity, as well as other non-traditional organoid/tumor constructs that maintain heterogeneity. This should be acknowledged.

Response: We have updated the statements according to the latest literature. We have now revised the relevant text (line 77-87) in the manuscript as follows:

“However, conventional cancer organoid models **mainly** represent tumor epithelium, and the endogenous stromal and immune cells have been gradually lost over time in culture.¹²⁻¹⁵ Although some studies reconstituted a part of TME in organoid culture systems by the air-liquid interface (ALI) method¹⁶ or co-culturing organoids and TME cells such as CAFs^{17,18} and immune cells^{19,20}, the models lacked controllability and uniformity in addition to labor cost. Some other cancer organoids derived from minced tumor fragments could maintain the native tissue architecture and TME cell components. However, manual tissue mincing results in non-reproducible fragment sizes and non-uniform environments^{21,22}. On the other hand, a limited number of millimeter-scale tumor fragments derived from small tumor tissues are limited in application in high-throughput drug screening.”

3. Introduction, line 86-89. It is not very challenging to use extrusion bioprinting to create small volume tumor constructs. 10 uL or even 5 uL is quite straightforward with most of today’s 3D bioprinters. Moreover, this approach can be use in a medium-to-high-throughput manner for rapid generation of tumor models. This is counter to the authors’ statements.

Response: We understand the reviewer’s concern that it is not very challenging to use extrusion bioprinting to create small volume tumor constructs nowadays. However, we want to emphasize that our research focuses on precisely manipulating microscale samples (e.g., 10 μ L bioink) to create approximately 200 LCAs with a volume of 0.05 μ L per LCA (\sim 400 μ m in diameter) in a fast, efficient method. Our goal is not to create tumor constructs of 5-10 μ L. We apologize for any confusion in our initial description, and we appreciate the opportunity to clarify our intentions.

Besides, we conducted further research to verify our claims. We found that existing 3D bioprinting techniques were capable of manipulating microsamples precisely. However, few studies have shown that 400 μ m-in-diameter tumor spheroids of $<$ 0.1 μ L in volume can be fabricated rapidly with good intra-batch consistency using 3D bioprinters.

4. Figure 1 is missing labels (e.g. (a), (b), etc.).

Response: Thank the reviewer for your careful review. We have modified the labels

accordingly.

5. The microinjection method seems unnecessarily complicate. Since the assembloids are simply organoids and other cells mixed together that form essentially a heterogeneous spheroid, couldn't "assembloids" be formed by hanging drop culture or culture in ultralow adhesion round bottom wells? Or by 3D printing small hydrogel droplets and crosslinking those?

Response: We appreciate the reviewer's suggestion. Here, we compared different methods (i.e. drop-let microfluidics, ultralow adhesion round bottom wells (ULA) and extrusion 3D bioprinting) to prove the necessity of the microinjection method when fabricating our LCAs.

1) Homogeneity and good morphological characteristics:

Droplet-based microfluidic systems could manipulate microvolume fluid and generate sub-microliter droplets in a controlled manner²³. In our manuscript, even 10 μL of cell-laden hydrogel could be manipulated precisely to produce ≥ 200 LCAs with a volume of 0.05 μL within 1 min by droplet microfluidic method (Fig 2j), enabling the drug screening for personalized medicine, especially for microvolume samples. More importantly, our LCAs generated by droplet microfluidics demonstrated a more uniform shape and cell distribution than those generated by co-culture in ULA microplates, which are often used to generate 3D spheroid tumor models (Response Fig. 1.1a). When using extrusion 3D-printing method, we can only form hemispheres of larger size with diameter exceeding 1mm, which is consistent with previous studies^{5,24}. Additionally, the cells within the hemispheres are selectively grown adherently, resulting in variability and imprecision in later drug screening (Response Fig. 1.1b). Although a 3D Bio-Dot printing method was developed for 3D spheroid formation, by which the cell-laden sacrificial bio-ink was bio-dot printed into a support hydrogel, one spheroid needed dispensing time of 3–9s. More importantly, the spheroids did not show the intra-batch consistency with uncontrollable growth and expansion in support hydrogel²⁵. The formed assembloids/microtissues via hanging drop culture exhibit various sizes and morphologies^{26,27}, making it challenging to achieve uniformity and reproducibility²⁸.

2) Reconstruction of microenvironment:

Droplet microfluidic method allows for the precise control of the chemical, biophysical and biological payload of each droplet²³. The optimized hydrogel provided biomimetic extracellular matrix environment with controllable mechanical properties (Fig. 2d). The assembloids generated by our droplet microfluidic method maintained a better TME of

the parental tumors, and the TME cells within our LCAs maintained better intra-batch consistency (Response Fig. 1.1c). A lack of the biomimetic extracellular matrix and precise control of the cell distribution may cause few TME cells existed in the assembloids formed by ULA method. Although the same density of cells and the same hydrogel were used, the tumor cells and TME cells of 3D printing group did not form tissue like morphology within the same culture time. One possible explanation for this divergence is weakened cell-cell interactions due to the stretching of the hydrogel on the surface of plates. Although a kidney organoid model with tissue morphology could be well fabricated by using extrusion-based 3D printing method, only 18 organoids with a diameter of ~2mm (0.55 μ L in volume for each organoid), could be generated per minute²⁴. Besides, the hanging drop culture method, much like the ULA method, promotes cellular aggregation without biomimetic extracellular matrix hydrogel²⁹, limiting the ability to construct the complex structures with TME.

In conclusion, the microinjection method based on droplet microfluidics offer better control over size and cell distribution, resulting in more uniform assembloids with defined morphological characteristics. Besides, the precise control of microfluidic system results in the creation of complex TME as well as the precise manipulation of clinical microvolume samples. These advantages highlight the need for microinjection method based on droplet microfluidics to achieve greater control and reproducibility in the formation of assembloids compared to other methods that the reviewer mentioned.

Response Fig. 1.1 (a) Morphological comparisons of assembloids formed by droplet microfluidic technology and ultralow adhesion round bottom wells (ULA). The average cell-seeding number is 2000 per well which is equivalent to the number of cells in a drop-let assembloid. (b) Morphological comparisons of assembloids formed by droplet microfluidic technology and extrusion 3D printing. (c) Immunofluorescence staining of human α -SMA, EpCAM and CD3 in LCAs fabricated by different methods. Scale bar, yellow bar, 200 μ m, white bar, 500 μ m.

6. It is certainly true that Matrigel is a potent matrix for supporting tumor cells and generating patient-derived organoids. However, it also is undefined and can essentially introduce variables that unintentionally can bias experimental results. Additionally, it is derived from a murine sarcoma tumor, which is inherently different than lung tumors – and most other tumors. The authors should acknowledge these problems.

Response: We agree with the reviewer and have illustrated the limitations of Matrigel for cancer organoid culture in Discussion section in revised manuscript (line463-466).

“Nevertheless, we have to admit the shortcomings of Matrigel-dependent hydrogel for cell or organoid culture. Matrigel is complex and poorly defined. Its complexity and batch-to-batch variation may bias cell selection and phenotype^{21 2}. We will develop

some alternative Matrigel-free materials for the generation of LCAs in the future.”

7. Mechanical analysis results: It should be stated in the results what kind of mechanical analysis was performed. Rheology? Indentation? Compression?

Response: We have stated it in the result and method sections: Compressive mechanical analysis was performed.

8. Results, lines 136-138. While technically correct, this is a bit of an overstatement. Using bioprinters that have been on the market for 5 years already, several hundred hydrogel-based tumor constructs can be created in several minutes. The microfluidic method is not significantly faster.

Response: We thank the reviewer for this comment, and we have changed the statement “Even 10 μL hydrogels containing $10^6\sim 10^8$ cells mL^{-1} could be successfully manipulated to generate about 200 uniform LCAs within 1 minute (Fig. 2i, Supplementary Fig.3g, h), which is not possible for both extrusion 3D bioprinting technology and manual operation.²⁶” to “ Even 10 μL hydrogels containing $10^6\sim 10^8$ cells mL^{-1} could be successfully manipulated to generate approximately 200 uniform LCAs with sizes of 400-500 μm ($\sim 0.05\mu\text{L}$ per LCA) within 1 minute (Fig. 2i, Supplementary Fig. 3g, h)”.

9. Other in vitro 3D tumor models are also bankable, so this is not a unique feature.

Response: We agree with the reviewer’s comment. Storable feature is important for tumor models. Therefore, we also verified that our LCA model could be cryopreserved for recovery and could be stored for further studies in our manuscript.

10. Figure 3: In panel b, some of the LCAs look very acellular. Others look highly irregular. This goes against the “consistency” touted by the authors.

Response: We acknowledge the reviewer’s doubt, and we would like to clarify that IHC staining for the markers were performed using the thin sections of paraffin embedded LCAs with a thickness of 5 μm . The cell numbers and cell distribution patterns are variable among different slices. Therefore, the IHC staining images appeared “inconsistent”.

More importantly, tumor samples derived from patients are heterogeneous, including but not limited to the spatial growth rate and cell morphology. Our LCAs not only

reflected this heterogeneity of parental tumors in morphology and histology but also maintained the consistency with similar shapes among intra-batch LCAs (Response Fig. 1.2a, b, Supplementary Fig. 4a). For instance, in LC05 tumor sample, tumor cells (CK7+/EpCAM+) were surrounded by a mass of extracellular matrices and CAFs (Response Figure 1.2b), which was also viewed in the corresponding LCA05 samples. CAFs stretched and grew around the tumor organoids, and tumor organoids were scattered in LCAs (Response Fig. 1.2a, Supplementary Fig. 4a). Additionally, CAFs with spindle shapes mainly containing cytoplasmic components only displayed the nucleuses, causing the “acellular” appearance in these IHC images of both LCAs and parental tumors.

However, in the sample derived from LC14, tumor cells were highly proliferative (high Ki67+ occupancy) with dense distribution (Supplementary Fig. 4b), which was also observed in corresponding LCA14 samples. Tumor cells grew rapidly in the 3-dimensional hydrogel and break through the spheroid boundaries to form less regular tumor-like tissue blocks, thus causing the less regular morphology of LCAs compared to that of LCA precursors with round shapes.

The histology heterogeneity was further validated in LC63 sample. LCAs of this sample could also form tissue-like morphology with protruding adenocarcinoma structures. Importantly, the highly differentiated morphological features of LC63 tumor were well maintained in LCAs, further supporting the conclusion of this part in our manuscript.

Together, these results demonstrated that LCAs effectively retained histological features of corresponding parental tumors as well as the intra-batch consistency.

Response Fig. 1.2 (a) Representative bright field microscopy images of LCAs derived from

LC05, LC14 and LC63 patients. (b) Sample images of immunohistochemistry staining for different markers showing the maintenance of the histopathological features of parental tumors in LCAs. Scale bar, 200 μ m.

11. Supplementary 4: Panel b – It's not clear if this is a fair comparison. There appears to be a lot more cellular material in the U-plate image than in the individual droplets. Moreover, in panel c, the U-plate merged panel doesn't look all that different than the LCAs.

Response: We sincerely appreciate the reviewer's insightful comments and constructive feedback on our manuscript. We acknowledge the importance of accurately balancing the cell numbers between the two methods used in our study. To address this concern, we have now made necessary adjustments to ensure that each type of assembly contains an equal number of cells at the beginning. Specifically, we have seeded an average of 2000 cells per assembloid in both droplet-based LCAs and ULA-based LCAs. After five days of culture, the assembloids were collected for further characterization. We have thoroughly investigated the batch-to-batch consistency between droplet-based LCAs and ULA-based LCAs. Our results demonstrated that droplet-based LCAs exhibit better consistency in terms of individual morphology and size, whereas ULA-based LCAs displayed more varied morphology and size distributions (Revised Supplementary Fig. 4f).

The immunofluorescence staining results indicated that droplet-based LCAs provide a more favorable environment for maintaining immune microenvironment cells and cancer-associated fibroblasts (CAFs), with consistent proportions across batches. In contrast, ULA-based LCAs predominantly consisted mainly of rapidly proliferating tumor cells, with a significantly lower proportion of tumor microenvironment cells, and a notable variation in composition within the same batch (Revised Supplementary Fig. 4g). One plausible explanation for this observation is that our hydrogel system in droplet-based LCAs provided a biomimetic extracellular matrix-like environment, promoting the cell growth. In contrast, the ULA-based method lacked such an exogenous matrix-like environment, requiring cells to secrete and gradually form their own extracellular matrix components, which posed challenges to cell growth and proliferation³⁰.

Revised Supplementary. Fig. 4 (f) Morphological comparisons of assembloids formed by droplet microfluidic technology and ultralow adhesion round bottom wells (ULA). The average cell-seeding number is 2000 per well which is equivalent to the number of cells in a drop-let assembloid. (g) Immunofluorescence staining of human α -SMA, EpCAM and CD3 in LCA55. Scale bar, yellow bar, 200 μ m.

12. RNAseq data: Since the genetic shift occurs over time like in organoids, how are the LCAs better than organoids?

Response: It is indeed acknowledged that long-term culture in vitro causes possible alterations of genetic and transcriptomic features of cancer models including two-dimensional cultured cells, cancer organoids^{6,31} and our own LCA model. This is a common challenge for in vitro cancer models caused by the culture systems and clonal selection. Nevertheless, our LCAs show the following advantages:

- 1) **Maintenance of heterogeneity and microenvironment.** LCAs were fabricated with droplet microfluidic technology using LCOs and TME cells. LCAs maintained the cancer epithelium heterogeneity, endogenous tumor microenvironments and cell-cell interactions of parental tumors, which were directly proven by the single cell RNA-seq results (Fig. 5a-f) and specific marker staining assays (Fig. 3c,d). LCAs could overcome the critical limitation of conventional lung cancer organoids with the lack of endogenous tumor-associated stromal components and immune cells^{32,33,6}.

2) **Intra-batch consistency and drug response profiling.** LCAs fabricated with the droplet microfluidic technology showed good intra-batch consistency in size and cell composition (Fig 2h, i), and LCAs presented better consistency in drug response profiling compared to the corresponding cancer organoids (Fig. 7c; Suppl. Fig 9a, b). In addition, LCAs could reconstruct functionally heterogeneous CAFs and immune environment of patients, enabling accurate evaluation of drug screening for combination therapy.

13. Figure 6. The patient comparison in panels e-g is nice. However, LCOs have TIL here, making them almost cellularly on par with the LCAs. The thought is that the TME adds resistance to the LCAs. Which part of the TME?

Response: The most critical difference between LCAs and LCOs in this treatment assay was that CAFs occurred in LCAs. To analyze whether CAFs derived from LC37 patient involved into the cancer cell protection from the drug treatments, we investigated the secretion levels of HGF and FGF7, which could identify the functional types of CAFs³⁴. The results demonstrated that CAFs of LC37 showed high secretion levels of HGF and FGF-7 (Revised Fig. 6a, b), suggesting its robustly protective roles of cancer cells from various treatments³⁴⁻³⁶.

Revised Fig. 6 (a, b) HGF and FGF7 secretion levels of CAFs derived from 6 different patients' tumors via enzyme linked immunosorbent assay (ELISA) (n = 3-5).

14. The data has LC's numbered up to LC42, yet only a selection are included in the presented data. How many samples were collected and how many generated successful LCOs/LCAs? Out of the successfully established LCOs/LCAs, how many recapitulated patient clinical results. Without these numbers it's impossible to assess whether this system was indeed successful, or if it was only predictive in 3 patients.

Response: We sincerely appreciate the reviewer for this valuable comment. The collected clinical samples in this manuscript were numbered according to the time series, some of which used for developing the system at the beginning were not listed in this manuscript. As shown in the revised Supplementary Table 1, we totally collected 49 clinical tumor samples including 43 adenocarcinoma samples and 6 squamous cell

carcinoma samples. LCAs were successfully established from 35 out of the 36 patient samples with a success ratio of 97.2% (Revised Supplementary table 1). Notably, the only case of failure to generate LCA was due to the presence of residual Matrigel during the LCO culture, which caused unsuccessfully photocrosslinked when fabricating LCA using droplet microfluidic method.

Among the 35 patients with successful LCAs, a total of 8 samples with clinical treatments were used to evaluate the drug response consistency between LCAs and patients, covering chemotherapy, targeting therapy, and combination immunotherapy. The results demonstrated that patient-specific LCAs revealed an overall accuracy of 88% in predicting their clinical outcomes (7 of 8) (Revised Fig.7m, Suppl.Fig.10). Of note, patient LC64 received neoadjuvant chemotherapy with a combination treatment of anti-PD-1 antibody plus Pemetrexed and Carboplatin for two cycles. Positron emission tomography-computed tomography (PET-CT) indicated that the patient had SD, which was inconsistent with the LCA prediction (Revised suppl. Fig. 10d). However, histological examination supported that the patient had significant pathological response with very few EpCAM⁺ tumor cells within the surgical tumor tissue (Revised suppl. Fig. 10e). Thus, after the correction by pathological response, the overall prediction accuracy of LCAs for the drug effectiveness reached 100% (8 of 8).

Revised supplementary Fig. 10d-h.

Revised Fig. 7m

15. The gelatin-methacrylate is photopolymerized, but the Matrigel is in a flowable form. How do the authors know that the Matrigel components don't slowly diffuse out of the assembloids. Validation of this should be included.

Response: We appreciate the reviewer's suggestion. Indeed, Matrigel could not be directly crosslinked and may diffuse out of the assembloids. To validate it, we fabricated cell-free microgels with or without Matrigel (Matrigel-, Matrigel+) by droplet

microfluidics and incubated them in PBS. The laminin that is the primary components of Matrigel³⁷, in the supernatant of each group were detected every day via using anti-mouse Laminin ELISA kit. As the result showed, one day after incubating, Matrigel started to slightly diffuse out into the medium. It exhibited significant increase of laminin in the incubated medium over time (Response Fig. 1.3a), indicating the gradual diffusion of Matrigel from the microgels.

Although Matrigel diffused gradually, the residual or diffused materials of Matrigel in assembloids contributed to the activity and proliferation of the cells within LCAs compare to that of Matrigel-free LCAs (Suppl. Fig 3b-d).

Response Fig. 1.3 (a) Matrigel diffusion analysis of cell-free gelMA-Matrigel microgel via enzyme linked immunosorbent assay (ELISA) for Laminin. ($n = 3$).

Reviewer #2 (Expertise: cancer genomics, Remarks to the Author):

In the manuscript entitled “A patient-specific lung cancer assembloid model with heterogeneous tumor microenvironments”, Zhang et al. proposed a droplet based microfluidic technology to generate lung cancer assembloids (LCA). With the ability to mimic the tumor microenvironment of in vivo tumors, such in vitro cancer models seems to be very promising for the development of personalized therapeutic approach. The authors validated how LCAs recapitulated actual tumors, in terms of the genomic/transcriptomic landscapes, TME composition etc. Overall, I do think the manuscript present an exciting advancement, I have a few comments in terms of improving the analysis and presentation.

Response: We sincerely appreciate the reviewer’s inspiring and valuable comments on our work. The following are point-by-point responses for the comments.

Major:

1. I think it would be helpful if the authors could mention some basic parameters of the

LCA (such as the mean diameter and the mean number cells involved) at the beginning of the text, even though numbers are already scattered around in figures and in the discussion.

Response: We appreciate the reviewer’s valuable suggestion. We have improved our description in the manuscript according to your suggestion as follows:

“The flow rates of the bioink and oil phase were optimized at 1 mL h⁻¹ and 5 mL h⁻¹ respectively to form uniform assembloid precursors with a diameter of 400-500 μm encapsulating a certain number of cells (e.g., 1500-2500 cells).”

2. Is it possible perform single-cell RNA-seq for cells in the LCAs? It seems to be very helpful to profile the cellular composition of LCAs. If not, what are the challenges? The authors should at least discuss that in the manuscript.

Revised Fig. 5. Single-Cell RNA-Seq analyses of parental tumors and corresponding LCAs.

(a) UMAP plot showing the coarsened clustering results colored by 6 major cell types. (b) UMAP plot showing the fine clustering results colored by 13 cell types. (c) UMAP plot showing all cells colored by sample IDs. (d) Comparative analysis between primary tumors (LC55 and LC66) and corresponding LCAs for the proportions of individual cell types. (e) Pearson correlation coefficients (PCC) obtained using cell type-specific differentially expressed genes.

Response: We appreciate the reviewer’s valuable suggestion. We have performed single-cell RNA sequencing (scRNA-seq) analysis of parental tumors from 2 patients and corresponding LCAs cultured for 1 and 2 weeks. We first integrated all cells from six datasets and identified six major cell types including epithelial cells, fibroblasts, T

cells, B cells, mast cells and macrophages (Revised Fig 5a, see Method section). Based on markers shown in Suppl. Fig. 6a, the epithelial cells could be clustered into 6 subtypes (e.g., Epithelium basal cells, AT2-like cells, AT2 cells, club cells and the cells in a proliferating state), and the fibroblasts cells and T cells were both clustered into 2 subtypes (non-proliferating and proliferating cells), suggesting the tumor and TME heterogeneity existed in these samples. LC55 and LC66 showed cell clustering similarity in shared major cell types except epithelial cells (basal and club cells were enriched in LC55 while AT2-like and AT2 cells were enriched in LC66) (Revised Fig 5c, Suppl. Fig. 6b), suggesting great patient heterogeneity in epithelial cells. Notably, all the cell types of patient tumors were maintained well in the corresponding LCAs and the proportions of cell types in 1-week-old LCAs were similar to those in parental tumors, whereas those in 2-week-old LCAs showed less similarity to parental tumors (Revised Fig. 5d, Suppl. Fig. 6c). Further investigation of transcriptome similarities between LCAs and their parental tumors showed that 1-week-old and 2-week-old LCAs displayed approximately 88% and 84% overall similarity to parental tumors, respectively. The epithelial and fibroblast cells of LCAs showed a slight decrease of overall similarity (92% to 86% in epithelial cells; 91% to 87% in fibroblast cells) to those of parental tumors over time. Of note, T cells of both 1-week-old and 2-week-old LCAs displayed high overall similarity to parental tumors with 93% and 92% similarity, respectively (Revised Fig. 5e, Suppl. Fig. 6d).

Together, single-cell RNA-seq analyses highlighted marked cellular heterogeneity in LCAs and further supported that LCAs recapitulate cell-type heterogeneity and molecular properties of corresponding parental tumors.

3. Even though single-cell transcriptomic profiling could be a challenge, it is still possible to shed light on cellular composition. The authors could find the expression profiles of CAFs in lung cancer as reference, and then perform deconvolution on their bulk RNA-seq experiments

Response: We have performed single-cell RNA-seq and profiled the cellular compositions of LCAs as shown in Revised Fig. 5 and Supplementary Fig. 6.

4. The effects of TME on tumors are mostly mediated by ligand-receptor interactions. Using curated ligand-receptor pairs, the authors can examine if the interactions are preserved from patient tumor to LCA using their bulk RNA-seq data. It could provide better understanding on the roles of CAFs in the models, as compared to the patient tumors.

Response: We thank the reviewer for the valuable suggestions that help us to improve our manuscript. To investigate the maintenance of cell-cell communications in LCAs,

ligand-receptor interaction analysis across all cell types was performed using single-cell RNA-seq data. The results showed that ligand-receptor interactions exhibited between any pair of two types of cell populations among fibroblast, epithelial cells, macrophage, T cells, B cells and mast cells in all samples. Both 1-week-old LCAs and 2-week-old LCAs maintained these cell-cell interactions (Revised Fig. 5f, Suppl. Fig. 7a). Moreover, the overall similarity of ligand-receptor interactions of CAFs-epithelial cells and Epithelial-CAFs between LCAs and parental tumors were about 78% and 68%, respectively, suggesting the maintenance of communications between tumor cells and CAFs in LCAs (Revised Fig. 5g) (The fibroblast cells derived from tumors in the figure is CAFs). The similarities of ligand-receptor interactions from CAFs to epithelial cells did not change significantly over time (76.8% to 78.8%), while a slight decrease of similarities was observed in the ligand-receptor interactions from epithelial cells to CAFs of LC55 sample (73% to 63%) (Revised Fig. 5g, h), consistent with that in LC66 (Suppl. Fig. 7b). And this may be caused by the changes in cell proportions and expression levels of ligand or receptor genes over time.

Of note, the ligand-receptor pairs of *HGF-MET* and *FGF-FGFR1* of fibroblast-epithelial cells and *TGFβ-(TGFBR1+TGFBR2)* of epithelial-fibroblast cells were observed in both parental tumors and LCAs (Revised Fig. 5g, h, Suppl. Fig. 7c), which are associated with functional heterogeneity of CAFs³⁴.

Together, single-cell RNA-seq analyses proved the maintenance of cell-cell communications of parental tumors in LCAs.

Revised Fig. 5 (f) Circle plot of cell-cell communication network of all cells. The width of edges represents the strength of the communication. (g) Venn diagram showing the overlap in ligand-receptor pairs from CAFs to epithelial cells between the LCAs and the parental tumor of LC55 patient. The ligand-receptor pairs of interest are listed in box. (h) Venn diagram showing the overlap in ligand-receptor pairs of Epithelial-CAFs cells between the LCAs and the parental tumor of LC55 patient. The ligand-receptor pairs of interest are listed in box.

5. An alternative cancer model that could capture the effects of TME is patient-derived xenograph, can the authors discuss compare LCA and PDX in terms of their usage as cancer models?

Response: Thank the reviewer for your insightful comments. We compared PDX and our LCAs models as follows:

1) Advantages and limitations of PDX models: PDX models are widely used in cancer research due to their ability to mimic patient-specific tumors. However, PDX models are not only time-consuming and expensive, they highly rely on the immunocompromised and humanized mice which exist ethical issues. Additionally, the engraftment and propagation of human tumor tissue in immunocompromised mice can lead to clonal selection pressures, resulting in genetic and phenotypic divergence from the parent tumor^{38,39}. Additionally, the stromal cells in PDX tumors are derived from murine origin, which are unable to reproduce personalized tumor microenvironments⁴⁰.

2) Advantages of LCA Models over PDX Models: Our LCA models offer several distinct advantages over PDX models. Firstly, LCA models can be quickly constructed, allowing for direct micro-modeling of clinical samples. Within approximately one week of in-vitro culture, LCA models can be utilized for drug screening, enabling personalized drug screening and guidance within two weeks. This significantly reduces time and economic costs, offering a crucial advantage in rapidly providing tailored treatment options for late-stage patients. Secondly, LCA models can recapitulate patient-specific tumor heterogeneity and TME, maintaining elements of the immune microenvironment of parental tumors, such as T cells, B cells, and macrophages, as well as functional heterogeneity in cancer-associated fibroblasts (CAFs). This critical feature enables the study of tumor-immune interactions and their influence on treatment response, a capability that is limited in PDX models.

In conclusion, LCA models represent a cost-effective, biologically relevant, and highly accurate platform in vitro for cancer research and drug screening, highlighting the potential of LCAs as an efficient and predictive approach for personalized cancer treatment.

6. One important feature of the LCA model claimed by the authors is that this model could maintain the cell composition of parental tumors. However, the authors only checked a few cell type-specific markers in a few samples. Considering the preparation of bioink, it's expected to see the markers shown in Fig. 1. 1) What are the patterns in other samples? Are they similar with those shown in Fig.1? 2) What are the proportions of tumor cells, CAFs and TILs in parental tumors and LACs? Single-cell RNA-seq may help.

Response: Following the reviewer’s suggestion, we have performed single-cell RNA-seq and compared the cellular compositions between LCAs and their parental tumors as shown in Revised Fig. 5 and Supplementary Fig. 6. LCAs recapitulated the inter- and intratumor heterogeneity of parental tumors and all the major cell types of patient tumors such as heterogeneous epithelial cells, fibroblast, T cells and B cells were maintained well in the corresponding LCAs. The results highlight marked cellular heterogeneity in LCAs and further support that LCAs recapitulate tumor heterogeneity and TME of corresponding parental tumors as shown in Fig.1.

The proportions of distinct cell types of LCAs and parental tumors were showed in Revised Suppl. Fig.6c and Response table 1. Overall, the proportions of distinct cell types in 1-week-old LCAs were more similar to those in parental tumors, compared to those in 2-week-old LCAs (Revised Fig. 5d, Revised Suppl. Fig.6c).

Samples	LC55			LC66		
	Tumor	LCA-1W	LCA-2W	Tumor	LCA-1W	LCA-2W
Epithelial	26.67%	54.25%	57.74%	80.09%	80.97%	61.33%
Fibroblast	45.41%	22.53%	18.19%	2.67%	1.05%	1.54%
T	25.82%	22.66%	23.16%	11.48%	11.93%	28.15%
B	0.66%	0.16%	0.50%	3.17%	2.74%	2.92%
Mast	0.82%	0.31%	0.10%	1.22%	1.98%	3.47%
Macrophage	0.62%	0.09%	0.31%	1.37%	1.33%	2.60%

Response Table 1: Cell proportions

7. One of the most significant drawbacks of conventional cancer organoid models, as mentioned by the authors at Line 72-74, is that these models can only represent tumor epithelium, and they loss stromal and immune cells gradually over time in culture. However, the changes over time in culture were also observed in LCA model (e.g. Fig. 4b). It’s of great significance to investigate more on these over-time changes of the LCA model: 1) how long could the main features (gene expression/cell composition/CAF heterogeneity/performance of drug test) be maintained? 2) how do these features change over time in culture? This should definitely be mentioned in the Discussion.

Response: We appreciate the reviewer’s valuable comments. To verify the effect of culture time on characteristics of LCAs, we performed new experiments following the reviewer’s suggestions.

To study LCA features of gene expression, cell composition and CAF heterogeneity over time, scRNA-seq analysis of parental tumors from 2 patients and corresponding LCAs cultured for 1 and 2 weeks was performed. All the cell types of patient tumors

were maintained well in the corresponding LCAs. The proportions of distinct cell types including CAFs in 1-week-old LCAs were similar to those in parental tumors, whereas those in 2-week-old LCAs showed a lower degree of similarity to parental tumors (Response Fig. 2.1a), which was further confirmed by the IF staining for CAFs (Response Fig. 2.1b).

The expression levels of cell type-specific genes showed that 1-week-old and 2-week-old LCAs displayed approximately 88% and 84% overall similarity to parental tumors, respectively. The cell-specific genes of epithelial and fibroblast cells in LCAs showed a little decrease of overall similarity (92% to 86% in epithelial cells; 91% to 87% in fibroblast cells) compared to those of parental tumors over time. Of note, cell-specific genes of T cells in 1-week-old and 2-week-old LCAs displayed high overall similarity to parental tumors with 93% and 92% similarity, respectively (Response Fig. 2.1c, d).

To study the effect of culture time on drug-responses of LCAs, we performed drug test assays in LCAs of 3 patient samples (LCA63, LCA66 and LCA55) cultured for 1 to 3 weeks. The results demonstrated that the overall response trends of LCAs to targeted drugs (Alflutinib, Ensartinib and Furmonertinib) and some chemotherapy drugs (Carboplatin) within 3 weeks of culture were consistent. While LCAs cultured for three weeks in individual cases (e.g., LCA66 and LCA63) became more sensitive to the chemotherapy drugs such as Taxol and Cisplatin (Response Fig. 2.1e-g).

Together, these results above indicated that LCAs cultured for 2 weeks could recapitulate the cell-type heterogeneity and molecular properties of corresponding parental tumors. The drug response profiles of LCAs may vary after three-week culture, highlighting the best time window for drug testing using LCAs is within 2 weeks.

Response Fig. 2.1 (a) Comparative analysis between primary tumors (LC55 and LC66) and corresponding LCAs for the proportions of individual cell types. T, T cells; B, B cells; Prolif, Proliferation; AT2, Alveolar type 2. (b) Immunofluorescence staining of human α -SMA and EpCAM in tumor fragments and corresponding LCAs (LC55 & LC66) cultured for 1 week and 2 weeks. (c) Heatmap showing the expression levels of cell type-specific top 50 genes in 3 sample types (tumors and corresponding LCAs cultured for 1 week and 2 weeks). (d) Pearson correlation coefficients (PCC) obtained using cell type-specific differentially expressed genes. (e-g) Dose-response curves of tumor assembloids (LCA63, LC66 and LCA55) cultured for different time. ($n = 3$ biologically independent cells, data are presented as mean \pm SEM).

Minor:

1. Line 184-186: please provide the correlation coefficients to support the statement that the transcriptome profiles are similar between LAC and corresponding parental

tumors. It's not obvious in Fig. 4a.

Response: We have showed the correlation coefficient of each tumor-LCAs pair in Fig. 4a and provided the overall correlation coefficient in the revised manuscript as follows:

“Transcriptome-wide comparisons demonstrated the high similarity between LCAs and their corresponding parental tumors with an overall correlation coefficient of 0.86 (Fig. 4a).”

2. Line 190-193: the authors provided the top enriched pathways of the genes up-regulated in LCA and tumors compared to normal samples. While the more interesting and meaningful comparison would be between LCA and tumors. Please perform the pathway enrichment analysis with the up-regulated genes in LCA (v.s. normal) and parent tumor (v.s. normal), and see how similar they are.

Response: Thank the reviewer for raising this question. Following the reviewer's suggestion, we have performed the pathway enrichment analysis with the up-regulated genes in LCAs (vs. normal) and the parental tumors (vs. normal). In particular, we compared the similarity of the top 20 enriched pathways between each parental tumor and LCA. In the LC14 group, 80.56% of the enriched pathways were found in both tumor and LCA including mitotic cell cycle, extracellular matrix organization. In the LC28 group, all the enriched pathways were the same in the tumor and LCAs including extracellular matrix organization and cell cycle processes. In the LC51 group, 69.7% of the pathways were co-enriched in both tumor and LCA, such as cell adhesion, ion transport, epithelial cell differentiation and so on. In the LC52 group, 75% of the enriched pathways were same in tumor and LCA including immune response and cell cycle (Response Fig. 2.2a). What's more, the top 20 enriched pathways between all parental tumors and all LCAs were compared and the result demonstrated almost 100% similarity (Response Fig. 2.2b), further supporting that LCAs could maintain the transcriptomic signatures of parental tumors.

Response Fig. 2.2 (a) Bubble plots of the GO pathway enrichment analysis of up-regulated genes in LCAs (vs. normal) and parent tumors (vs. normal) of each sample. The top 20 enriched pathways in both LCAs and parent tumors were compared. The size of the bubble indicates the gene numbers annotated in a pathway term to all gene numbers annotated in this pathway term. A greater bubble indicates a higher degree of pathway enrichment. The color indicates the p values. (b) The comparison of the top 20 pathways enriched in both LCAs and parent tumors among of all the samples.

3. Fig. 1: the numbering of panels, a, b and c, are missing.

Response: We have modified the labels accordingly.

4. Fig. 4a and e: the color of the bars indicating Normal, Tumor and LCA should be consistent between LC14 and LC28.

Response: We have modified them accordingly.

5. Fig. 4f: how these 63 genes were selected?

Response: We analyzed the mutated genes in LCAs and their parental tumors separately, and then identified the shared mutational genes. These 63 genes were the shared mutational genes in LC28 tumor and its corresponding LCAs.

6. Fig. 4g: redundant with Fig. 4f. It might be better to just highlight these genes in Fig. 4f.

Response: We have increased the number of tested samples. The related data of Fig. 4f of the previous version has been included in Supplementary Figure 5e. The somatic variants in lung cancer-associated genes are displayed in Fig. 4f.

7. Fig. 4h: the heatmap needs to be properly scaled. Currently it's not intuitive to say the CNV profiles are similar between T and A. Can the authors comment on that?

Response: We acknowledge the Reviewer's doubt. It is indeed not appropriate to draw the above conclusions based on the data from only one sample. Therefore, we performed whole exome sequencing (WES) analysis in another two pairs of primary lung tumors, normal tissue, and matched LCAs. The results demonstrated that LCAs and parental tumors shared similar CNV profiles with an overall rate of 76% similarity (Fig. 4f, Suppl. Fig. 5f).

8. Line 273, the conclusion is not too obvious as seen in Supplementary Fig. 7a.

Response: The undesirable display of the differences may be caused by not scaling the heat maps properly. Therefore, we adjusted the scale of each heat map and the revised figure could better display the differences between the LCAs and LCOs groups (Suppl. Fig. 9a).

9. Fig. S7b: it might be more appropriate to use grouped bars instead of stacked bars.

Response: Thank the reviewer for the valuable suggestion. We have changed the attacked bars into grouped bars (Suppl. Fig. 9b).

References

- 1 Kleinman, H. K. & Martin, G. R. Matrigel: basement membrane matrix with biological activity. *Semin Cancer Biol* **15**, 378-386 (2005).
- 2 Kozłowski, M. T., Crook, C. J. & Ku, H. T. Towards organoid culture without Matrigel. *Commun Biol* **4**, 1387 (2021).
- 3 Lago, C. *et al.* Medulloblastoma and high-grade glioma organoids for drug screening, lineage tracing, co-culture and in vivo assay. *Nat Protoc* **18**, 2143-2180 (2023).
- 4 Sebastian, R. *et al.* Schizophrenia-associated NRXN1 deletions induce developmental-timing- and cell-type-specific vulnerabilities in human brain organoids. *Nat Commun* **14**, 3770 (2023).
- 5 Kim, E. *et al.* Creation of bladder assembloids mimicking tissue regeneration and cancer. *Nature* **588**, 664+ (2020).
- 6 Lo, Y. H., Karlsson, K. & Kuo, C. J. Applications of Organoids for Cancer Biology and Precision Medicine. *Nat Cancer* **1**, 761-773 (2020).
- 7 Zhang, Y. M. *et al.* 3D Bioprinted GelMA-Nanoclay Hydrogels Induce Colorectal Cancer Stem Cells Through Activating Wnt/beta-Catenin Signaling. *Small* **18** (2022).
- 8 Yue, K. *et al.* Synthesis, properties, and biomedical applications of gelatin methacryloyl (GelMA) hydrogels. *Biomaterials* **73**, 254-271 (2015).
- 9 Liu, T. T., Weng, W. X., Zhang, Y. Z., Sun, X. T. & Yang, H. Z. Applications of Gelatin Methacryloyl (GelMA) Hydrogels in Microfluidic Technique-Assisted Tissue Engineering. *Molecules* **25** (2020).
- 10 Deng, J. *et al.* Photocurable Hydrogel Substrate-Better Potential Substitute on Bone-Marrow-Derived Dendritic Cells Culturing. *Materials (Basel)* **15** (2022).
- 11 Zhang, Y. *et al.* 3D Bioprinted GelMA-Nanoclay Hydrogels Induce Colorectal Cancer Stem Cells Through Activating Wnt/beta-Catenin Signaling. *Small* **18**, e2200364 (2022).
- 12 Lo, Y. H., Karlsson, K. & Kuo, C. J. Applications of organoids for cancer biology and precision medicine. *Nat Cancer* **1**, 761-773 (2020).
- 13 LeSavage, B. L., Suhar, R. A., Broguiere, N., Lutolf, M. P. & Heilshorn, S. C. Next-generation cancer organoids. *Nat Mater* **21**, 143-159 (2022).
- 14 Hofer, M. & Lutolf, M. P. Engineering organoids. *Nat Rev Mater* **6**, 402-420 (2021).
- 15 Yin, S. Y. *et al.* Patient-derived tumor-like cell clusters for drug testing in cancer therapy. *Sci Transl Med* **12** (2020).
- 16 Neal, J. T. *et al.* Organoid Modeling of the Tumor Immune Microenvironment. *Cell* **175**, 1972-1988 e1916 (2018).
- 17 Naruse, M., Ishigamori, R., Ochiai, M., Ochiai, A. & Imai, T. Gene expression profiles in CAFs exhibited individual variations by the co-culture with CRC organoids. *Cancer Science* **113**, 345-345 (2022).
- 18 Liu, J. Y. *et al.* Cancer-Associated Fibroblasts Provide a Stromal Niche for Liver Cancer Organoids That Confers Trophic Effects and Therapy Resistance. *Cell Mol Gastroenter* **11**, 407-431 (2021).
- 19 Cattaneo, C. M. *et al.* Tumor organoid-T-cell coculture systems. *Nature Protocols* **15**, 15-39 (2020).
- 20 Zhang, Y. M. *et al.* 3D Bioprinted GelMA-Nanoclay Hydrogels Induce Colorectal Cancer Stem Cells Through Activating Wnt/beta-Catenin Signaling. *Small* (2022).
- 21 LeSavage, B. L., Suhar, R. A., Broguiere, N., Lutolf, M. P. & Heilshorn, S. C. Next-generation

- cancer organoids. *Nat Mater* **21**, 143-159 (2022).
- 22 Voabil, P. *et al.* An ex vivo tumor fragment platform to dissect response to PD-1 blockade in cancer. *Nat Med* **27**, 1250-1261 (2021).
- 23 Morneau, D. Droplet-based microfluidics. *Nature Reviews Methods Primers* **3** (2023).
- 24 Lawlor, K. T. *et al.* Cellular extrusion bioprinting improves kidney organoid reproducibility and conformation. *Nat Mater* **20**, 260-+ (2021).
- 25 Jeon, S., Heo, J. H., Kim, M. K., Jeong, W. & Kang, H. W. High-Precision 3D Bio-Dot Printing to Improve Paracrine Interaction between Multiple Types of Cell Spheroids. *Advanced Functional Materials* **30** (2020).
- 26 Fang, Y. & Eglén, R. M. Three-Dimensional Cell Cultures in Drug Discovery and Development. *SLAS Discov* **22**, 456-472 (2017).
- 27 Cho, C. Y. *et al.* Development of a Novel Hanging Drop Platform for Engineering Controllable 3D Microenvironments. *Front Cell Dev Biol* **8**, 327 (2020).
- 28 Stuart, T. *et al.* Comprehensive Integration of Single-Cell Data. *Cell* **177**, 1888-1902 e1821 (2019).
- 29 Shri, M., Agrawal, H., Rani, P., Singh, D. & Onteru, S. K. Hanging Drop, A Best Three-Dimensional (3D) Culture Method for Primary Buffalo and Sheep Hepatocytes. *Sci Rep-Uk* **7** (2017).
- 30 Weiswald, L. B., Bellet, D. & Dangles-Marie, V. Spherical Cancer Models in Tumor Biology. *Neoplasia* **17**, 1-15 (2015).
- 31 Jacob, F. *et al.* A Patient-Derived Glioblastoma Organoid Model and Biobank Recapitulates Inter- and Intra-tumoral Heterogeneity. *Cell* **180**, 188-204 e122 (2020).
- 32 Kim, M. *et al.* Patient-derived lung cancer organoids as in vitro cancer models for therapeutic screening. *Nature Communications* **10** (2019).
- 33 Hu, Y. W. *et al.* Lung cancer organoids analyzed on microwell arrays predict drug responses of patients within a week. *Nature Communications* **12** (2021).
- 34 Hu, H. *et al.* Three subtypes of lung cancer fibroblasts define distinct therapeutic paradigms. *Cancer Cell* **39**, 1531-1547 e1510 (2021).
- 35 Huang, X. *et al.* Targeting the HGF/MET Axis in Cancer Therapy: Challenges in Resistance and Opportunities for Improvement. *Frontiers in Cell and Developmental Biology* **8** (2020).
- 36 Raghav, K. P., Gonzalez-Angulo, A. M. & Blumenschein, G. R., Jr. Role of HGF/MET axis in resistance of lung cancer to contemporary management. *Transl Lung Cancer Res* **1**, 179-193 (2012).
- 37 Aisenbrey, E. A. & Murphy, W. L. Synthetic alternatives to Matrigel. *Nature Reviews Materials* **5**, 539-551 (2020).
- 38 Day, C. P., Merlino, G. & Van Dyke, T. Preclinical Mouse Cancer Models: A Maze of Opportunities and Challenges. *Cell* **163**, 39-53 (2015).
- 39 Ben-David, U. *et al.* Patient-derived xenografts undergo mouse-specific tumor evolution. *Cancer Research* **78** (2018).
- 40 Zhang, Y. M. *et al.* Establishing metastatic patient-derived xenograft model for colorectal cancer. *Japanese Journal of Clinical Oncology* **50**, 1108-1116 (2020).

REVIEWER COMMENTS

Reviewer #1 (Remarks to the Author):

In this revised manuscript, the authors provide a fairly comprehensive set of revisions that largely successfully respond to the reviewer comments.

In particular, the authors increased the number of patient-to-LCA drug response comparisons and were explicit about their success rate in generating viable LCAs from tumor biospecimens. These two items significantly increase the impact of the overall paper. In addition, care has been taken to improve grammar and the overall written quality. However, several items remain that could be further addressed.

Specific Comments:

1. The authors did somewhat address the previous comment that critiqued the original submission for stating that organoids only represent tumor epithelium and that other approaches to generating 3D tumor models can preserve tumor heterogeneity. There are non-Matrigel-based hydrogel 3D constructs comprised of heterogeneous patient-derived tumor cells. It would be great to include some reference to examples of these in order to be comprehensive.
2. The authors make a reasonable point in terms of using the Drop-let assembloids versus 3D printed assembloids. The images of 3D printed models included in Response Fig. 1.1. would be great to include in the Supplemental Materials.
3. The authors' response to the comment about the acellular nature of some of the LCAs and subsequent irregularity and inconsistency is well understood. The LCAs vary because the biology of the original tumors vary. However, this does not address the fact that the LCAs can be quite inconsistent. This limitation should be acknowledged. It will be perfectly ok to acknowledge this limitation, and then explain that it is due to the variations in the original tumors, which the platform relatively successfully supports.
4. The acknowledgement that alterations in genetic and transcriptomic features do indeed occur in the LCAs, like organoids, should be included in the text. In addition, as described above, maintenance of heterogeneous cell populations like stromal and immune cells is possible in other systems. The LCAs do not uniquely do this. As such the claims that they do this with superiority over other model systems should be toned down.
5. Regarding Figure 6, and the authors stating that cancer associated fibroblasts (CAFs) may be the key difference maker in the LCAs compared to LCOs – why can't CAFs be included in organoid cultures?
6. Response Figure 1.3 is appreciated. However, this result should be acknowledged in the text. If laminin (along with other materials most likely) is slowly diffusing out of the LCAs, over time, the material composition will change, and will likely diminish in terms of the LCA platform being able to support viability, proliferation, heterogeneity, etc. This is a limitation.

Reviewer #2 (Remarks to the Author):

The authors did a great job addressing my concerns.

Reviewer #3 (Remarks to the Author):

NCOMMS 22-46035B

Zhang et al. " A patient-specific lung cancer assembloid model with heterogeneous tumor microenvironments "

The authors present data on generating lung cancer "assembloids" (LCAs) using surgical resection specimens and in one or a few cases, small needle biopsy specimens from 36 non-small cell lung cancer patients that is based on using a microfluidic device they created and then mixing lung cancer organoids in Matrigel with cancer associated fibroblasts (CAFs) and components of the tumor microenvironment (TME). They claimed success in 35/36 patients and showed these LCAs could be biobanked (frozen and thawed) for subsequent studies. They then studied the LCAs for comparison to the tumors from which they were derived by histology and molecular analyses (RNA expression, mutation), determination of scRNAseq for a couple, drug response phenotypes, for a subset, and comparison of therapy response phenotypes from the responses seen in patients in a small subset. They conclude: " This method enables precise manipulation of clinical microsamples and rapid generation of LCAs with good intrabatch consistency in size and cell composition. LCAs recapitulate the inter- and intratumoral heterogeneity, TME cellular diversity, and genomic and transcriptomic landscape of their parental tumors. This LCA model could help us identify the functional heterogeneity of cancer associated fibroblasts that correlates with tumor responses to drug treatments. More importantly, the LCA model presents better consistency in drug response profiling compared to the corresponding cancer organoids and accurately replicates the clinical outcomes of patients, suggesting the potential of the LCA model to predict personalized treatments. Collectively, our studies provide a valuable method for precisely fabricating cancer assembloids and a promising LCA model for cancer research and personalized medicine. "

Comments to the Authors:

This manuscript is reviewed in the context of the urgent need to have assays on patient lung cancers that would allow selection of the best chemotherapy, targeted therapy, immune checkpoint blockade therapy, and potential radiation therapy. It is also reviewed in the context of the large literature of prior reports attempting to do these same things ranging from the use of tumor cell lines, patient derived xenografts, and organoid cultures, and the requirement to have special equipment similar to that developed "in house" by the authors. In addition, this manuscript had been extensively reviewed previously and the authors have devoted 31 pages of detailed rebuttal to the prior comments including extensive editing and provision of additional data. However, their manuscript is still very, very complex and hard to get through to determine if they made the main points they can actually claim. A major part of this, have they provided evidence that the field should shift to working with LCAs vs. LCOs (or other short term cultures) for studying therapy response phenotypes both for use in selecting patient therapies and also, importantly as a preclinical model for testing new therapies including new immunotherapies and targeting the TME?

1. The authors have done an excellent job in their response to each of the very large number of general and specific comments by the two previous reviewers who were obviously expert in the area of this manuscript. Because of this I will restrict my comments to major issues that deal with presentation of the data and a tremendous need to make their presentation much simpler and straightforward. While they have done a very large amount of computational biology analyses, the way this is presented, makes it very difficult to determine what exactly they have shown and contributed to the field.
2. Their work is based on the "in house" microfluidics device they have constructed. Perhaps the information of how to make a similar device (schematics, other publications, component parts) is there, but not obvious. The first step is to provide detailed instructions of how someone would assemble a similar device. This could be supplemental, on line, but there needs to be step by step information how to do this. If a company is already making this and it is possible to purchase such a device we need to know.
3. A major point they make is comparison of the histology between the LCAs and the tumors they were derived from. While there are many ways to do this in a systematic way (e.g. blinded histology diagnoses of the tumors and LCAs into major lung cancer histologies and then see how they match) – perhaps the most straightforward thing would be to supply as supplemental data the electronic images of the tumors and the LCAs in H&E stains and the readers can then assess how well they match one another.
4. Another comparison they have provided is bulk RNA expression data of the tumor and the LCAs.

As part of this they need to simply state how many mRNA comparisons between different tumors and LCAs they have made for the current heat maps do not easily provide this information. One straightforward way would be to take published mRNA signatures or well known markers for different types of differentiation and see how these compare (e.g. lung adenocarcinoma, lung squamous cancers, large cell, neuroendocrine, mixed histologies). Another straightforward approach is to determine how closely tumors and LCAs from the same patient compare, versus how the LCAs compare to all other patient tumor samples. In addition, they may have done this, but if they compare RNA expression data from individual LCAs from the same patient how similar are they – and how many of these comparisons have they made?

5. They make a big point about the LCAs being more representative than lung cancer organoids (LCOs) developed in U shaped wells and or LCAs compare to LCOs from the same patient. They need to clearly state how many of these comparisons they have done, what was used for the comparison and how similar or different they are. It may be that LCAs compared to LCOs from the same patient are actually pretty similar for a variety of metrics (e.g. mRNA expression, mutations, freeze thaw stability, drug response phenotypes). Thus, even if LCAs may be better (a lot or only slightly) than LCOs derived by less fancy means, for most usage LCOs may be the easier way to go. Throughout the results they say they have made comparisons, it appears these may be only for a few samples, but some kind of summary of all of the comparisons including number tested for each metric and what was found and how different or similar they are would be useful to know.

6. As part of this the scRNAseq data are important. However, as far as I can tell this only done in a very few examples (2 patients I believe). While more scRNAseq data comparisons would increase the value of the manuscript, they need to start out early in the Results section and state how many comparisons were made, and what the results are – both for LCAs and the tumors, and for comparing LCAs with LCOs from the same patient.

7. Obviously a major point the authors make is the potential utility of the LCAs for determining ex vivo drug response phenotypes. What we need are LCA vs. LCO comparisons of drug response phenotypes for drugs commonly used in the treatment of lung cancer. While the IC50 and other values could differ between cells grown in the two methods, what we need to know are the relative responses of different regimens – the key data that would ultimately be used in selecting therapy for individual patients. Thus, let's say there are a few common chemotherapy regimens used for the treatment of non-small cell lung cancer based on platin doublets (e.g. platin + taxane, platin + gemcitabine, platin + pemetrexed) we need to know for LCAs from one patient: what is the reproducibility of drug response phenotype for each regimen; what are the relative ranks of sensitivity and resistance, and quantitatively how big are these differences? Then, we need to compare this to the same analyses in LCOs from the same patients. Do you get the same relative rankings, are the variations similar or different. Finally, if you look at relative rankings and the quantitative differences in response phenotypes between LCAs from different patients, how similar or different are they? This is the exact type of information one would need to use this assay in a clinical trial and from the current presentation it is not clear what the data would be or are.

8. They have made comparisons of the assays in the LCAs with patient responses for 8 patients (Supplementary Table 2). In most of these examples the therapy was given as neoadjuvant therapy (where response rates are very high, and in fact nearly all of the patients experienced what the authors scored as a "PR" with just one stable disease. Because of the complex nature of such comparisons and the fact that each patient was only treated with one regimen it is impossible to know if LCA guided therapy was actually working in this examples. To keep this simple, we need to know for each patient what were the LCAs results of various drug/targeted therapy, immune checkpoint blockade response phenotypes tested and how did they compare to one another was one regimen significantly different (more sensitive or resistant than an of the others? Also if you compare the results between the 8 patients were the ranking of sensitivity and resistance of the different regimens similar or different? As part of this we know from neoadjuvant studies of NSCLC that 60+ plus of tumors will have a partial response or better to chemotherapy combinations, a similar or higher response if they have EGFR mutations to TKI targeted therapy, and a similar high response to immune checkpoint blockade delivered as neoadjuvant therapy. What is important in terms of long term survival is the generation of a pathologic complete response (or near complete response after such neoadjuvant therapy) and here is where testing of LCA or LCO response phenotypes could help select therapy. Obviously such information will have to come from subsequent clinical trials.

RESPONSE TO REVIEWERS' COMMENTS

We would like to thank the reviewers for your careful reading, helpful comments, and constructive suggestions, which has significantly improved the presentation of our manuscript. In this revised version, changes to our manuscript within the document were all highlighted by using purple colored text. Here we present a detailed point-by-point response to the comments raised by the reviewers.

Reviewer #1 (Remarks to the Author):

In this revised manuscript, the authors provide a fairly comprehensive set of revisions that largely successfully respond to the reviewer comments.

In particular, the authors increased the number of patient-to-LCA drug response comparisons and were explicit about their success rate in generating viable LCAs from tumor biospecimens. These two items significantly increase the impact of the overall paper. In addition, care has been taken to improve grammar and the overall written quality. However, several items remain that could be further addressed.

Response: We appreciate the reviewer's valuable comments on our manuscript that we used to improve our work. We also appreciate the reviewer's affirmation of last round of revision and we have tried our best to address all the items you mentioned below.

Specific Comments:

1. The authors did somewhat address the previous comment that critiqued the original submission for stating that organoids only represent tumor epithelium and that other approaches to generating 3D tumor models can preserve tumor heterogeneity. There are non-Matrigel-based hydrogel 3D constructs comprised of heterogeneous patient-derived tumor cells. It would be great to include some reference to examples of these in order to be comprehensive.

Response: We appreciate the reviewer's valuable comments and we have included some references you suggested into the manuscript (Line 80-82) as follows:

“Although some studies reconstituted a part of the TME in organoid culture systems by the air-liquid interface (ALI) method¹ or coculturing organoids with TME cells such as CAFs^{2, 3} and immune cells^{4, 5}, some other studies developed non-Matrigel-based hydrogel 3D cancer models comprised of heterogeneous patient-derived tumor cells and stromal cells^{6, 7}, the models lacked precise controllability and uniformity in addition to labor costs. Some other cancer organoids derived from minced tumor fragments could maintain the native tissue architecture and TME cell components. However,

manual tissue mincing results in nonreproducible fragment sizes and nonuniform environments^{8, 9, 10}.”

2. The authors make a reasonable point in terms of using the Drop-let assembloids versus 3D printed assembloids. The images of 3D printed models included in Response Fig. 1.1. would be great to include in the Supplemental Materials.

Response: Many thanks for the reviewer’s suggestion and the images of 3D printed models in Response Fig. 1.1 have been included in the Supplementary Fig.4f-g.

3. The authors’ response to the comment about the acellular nature of some of the LCAs and subsequent irregularity and inconsistency is well understood. The LCAs vary because the biology of the original tumors vary. However, this does not address the fact that the LCAs can be quite inconsistent. This limitation should be acknowledged. It will be perfectly ok to acknowledge this limitation, and then explain that it is due to the variations in the original tumors, which the platform relatively successfully supports.

Response: We appreciate the comments and have revised the article accordingly (Line172-175).

“Tumors have the features of inter- and intratumor heterogeneity, including but not limited to cellular and histological heterogeneity^{41 42}. The LCAs derived from the same patients or different patients showed heterogeneous morphology, suggesting the maintenance of inter- and intratumor heterogeneity of patients (Supplementary Fig. 4a).”

4. The acknowledgement that alterations in genetic and transcriptomic features do indeed occur in the LCAs, like organoids, should be included in the text. In addition, as described above, maintenance of heterogeneous cell populations like stromal and immune cells is possible in other systems. The LCAs do not uniquely do this. As such the claims that they do this with superiority over other model systems should toned down.

Response: We appreciate the reviewer’s valuable comments and we have illustrated various cancer models including those systems contained TME in the **Instruction** Section (Line 78-82):

“**Although some studies reconstituted a part of the TME in organoid culture systems by the air-liquid interface (ALI) method¹ or coculturing organoids with TME cells such as CAFs^{2,3} and immune cells^{4,5}, some other studies developed non-Matrigel-based hydrogel 3D cancer models comprised of heterogeneous patient-derived tumor cells and stromal cells^{6,7}**, the models lacked precise controllability and uniformity in addition to labor costs. Some other cancer organoids derived from minced tumor fragments could maintain the native tissue architecture and TME cell components. However, manual tissue mincing results in nonreproducible fragment sizes and nonuniform environments^{8, 9, 10}.”

We have also explained the alterations in genetic and transcriptomic features in the manuscript (Line 211-214) as follows:

“However, the similarity in transcriptome decreased with the culture time of both cancer organoids and TME cells (the correlation coefficients of gene expression in LC28 and LC14 were 0.9 and 0.75, respectively) (Fig. 4a, Supplementary Fig. 5a, b), consistent with that in organoids¹¹.”

5. Regarding Figure 6, and the authors stating that cancer associated fibroblasts (CAFs) may be the key difference maker in the LCAs compared to LCOs – why can't CAFs be included in organoid cultures?

Response: When co-culturing LCOs and CAFs together in matrigel, CAFs were attached to the bottom of the dish (Response Fig. 1) in different dimensions with organoids seeded inside the Matrigel. However, in our LCA model, it was proved that CAFs and organoids were fully intermingled and fused with good interactions.

Response Fig. 1: (a) The images of co-culturing LCOs and CAFs derived from 4 different patients in matrigel. The LCOs were indicated by blue arrows and CAFs by red arrows. (b) Immunofluorescence staining of LCOs for the epithelial cell marker E-cadherin, CAFs marker FAP and endothelial marker CD31. **Few CAFs could be observed in LCOs.**

6. Response Figure 1.3 is appreciated. However, this result should be acknowledged in the text. If laminin (along with other materials most likely) is slowly diffusing out of the LCAs, over time, the material composition will change, and will likely diminish in terms of the LCA platform being able to support viability, proliferation, heterogeneity, etc. This is a limitation.

Response: We appreciate the comments and have illustrated the result in our manuscript (Line 478-480) accordingly.

“Nevertheless, we must admit the shortcomings of Matrigel-dependent hydrogels for cell or organoid culture. Matrigel is complex and poorly defined. Its complexity and lot-to-lot variation may bias cell selection and phenotype^{17 66}. In addition, **Matrigel exhibit gradual diffusion out of LCAs over culture time, leading to a potential decrease in consistency after a relatively long-term culture.** We will develop some alternative Matrigel-free materials for the generation of LCAs in the future.”

Reviewer #2 (Remarks to the Author):

The authors did a great job addressing my concerns.

Response: We highly appreciate your very helpful comments.

Reviewer #3 (Remarks to the Author):

NCOMMS 22-46035B

Zhang et al. “ A patient-specific lung cancer assembloid model with heterogeneous tumor microenvironments “

The authors present data on generating lung cancer “assembloids” (LCAs) using surgical resection specimens and in one or a few cases, small needle biopsy specimens from 36 non-small cell lung cancer patients that is based on using a microfluidic device they created and then mixing lung cancer organoids in Matrigel with cancer associated fibroblasts (CAFs) and components of the tumor microenvironment (TME). They claimed success in 35/36 patients and showed these LCAs could be biobanked (frozen and thawed) for subsequent studies. They then studied the LCAs for comparison to the tumors from which they were derived by histology and molecular analyses (RNA expression, mutation), determination of scRNAseq for a couple, drug response phenotypes, for a subset, and comparison of therapy response phenotypes from the responses seen in patients in a small subset. They conclude: “ This method enables precise manipulation of clinical microsamples and rapid generation of LCAs with good intrabatch consistency in size and cell composition. LCAs recapitulate the inter- and intratumoral heterogeneity, TME cellular diversity, and genomic and transcriptomic landscape of their parental tumors. This LCA model could help us identify the functional heterogeneity of cancer associated fibroblasts that correlates with tumor responses to drug treatments. More importantly, the LCA model presents better consistency in drug response profiling compared to the corresponding cancer organoids and accurately replicates the clinical outcomes of patients, suggesting the potential of the LCA model to predict personalized treatments. Collectively, our studies provide a valuable method for precisely fabricating cancer assembloids and a promising LCA model for cancer research and personalized medicine. “

Comments to the Authors:

This manuscript is reviewed in the context of the urgent need to have assays on patient lung cancers that would allow selection of the best chemotherapy, targeted therapy, immune checkpoint blockade therapy, and potential radiation therapy. It is also reviewed in the context of the large literature of prior reports attempting to do these same things ranging from the use of tumor cell lines, patient derived xenografts, and organoid cultures, and the requirement to have special equipment similar to that developed “in house” by the authors. In addition, this manuscript had been extensively reviewed previously and the authors have devoted 31 pages of detailed rebuttal to the prior comments including extensive editing and provision of additional data. However,

their manuscript is still very, very complex and hard to get through to determine if they made the main points they can actually claim. A major part of this, have they provided evidence that the field should shift to working with LCAs vs. LCOs (or other short term cultures) for studying therapy response phenotypes both for use in selecting patient therapies and also, importantly as a preclinical model for testing new therapies including new immunotherapies and targeting the TME?

Response: Many thanks for these valuable comments. In this study, we construct a tumor-like assembloid model (LCA) with tumor microenvironment based on droplet microfluidic technology with high consistency, in response to the lack of tumor microenvironment, which is a common problem in traditional models such as LCOs. Our LCAs recapitulate the inter- and intra-tumor heterogeneity, TME cellular diversity, and genomic and transcriptomic landscape of their parental tumors. More importantly, LCAs show better cell viability consistency among parallel experiments and personalized drug response characteristics in drug screening compared to LCOs. LCAs can show the effect of heterogeneous microenvironment such as CAFs on drug responses, while LCOs cannot.

We have revised several related paragraphs for the above points according to your suggestions. Hope this revision is much clearer.

1. The authors have done an excellent job in their response to each of the very large number of general and specific comments by the two previous reviewers who were obviously expert in the area of this manuscript. Because of this I will restrict my comments to major issues that deal with presentation of the data and a tremendous need to make their presentation much simpler and straightforward. While they have done a very large amount of computational biology analyses, the way this is presented, makes it very difficult to determine what exactly they have shown and contributed to the field.

Response: Thanks for these valuable comments and suggestions. Besides the conventional phenotypic validation, we also conducted RNA-seq, whole-exome sequencing and single-cell RNA-seq to verify the high similarity among tumors and LCAs in mRNA expression profiles, functional pathways, somatic mutations, copy number alteration, cell compositions of tumor microenvironments. The computational analyses of those molecular data are implemented to conduct more comprehensive comparisons. We have carefully revised the related sentences to better deliver this point according to your kind suggestions.

2. Their work is based on the “in house” microfluidics device they have constructed. Perhaps the information of how to make a similar device (schematics, other publications, component parts) is there, but not obvious. The first step is to provide detailed instructions of how someone would assemble a similar device. This could be supplemental, on line, but there needs to be step by step information how to do this. If a company is already making this and it is possible to purchase such a device we need to know.

Response: This assembly platform was built by us in response to our experimental requirements. At present, no company owns the device, and we are developing it into a more versatile commercial device.

Following the reviewer's suggestions, we have supplemented the methodology section with a detailed description of the platform construction process (Line550-567).

“Microfluidic Platform Setup: The microfluidic platform was set up based on our previous work¹². Briefly, the microfluidic platform mainly consisted of two infusion pumps, a T-junction PDMS chip, a lighting module, a microscope module, and a heating module. The T-junction PDMS chip was fabricated by soft lithography, connected with a piece of silicone tubing (Woer, ID = 0.56 mm) with a blunt end G22 needle. Both the hydrogel precursor and oil phases were loaded into the silicone tubes, which were separated by air bubbles and further pushed by the aqueous solution in the 1 mL plastic injection syringe controlled by the infusion pumps. Microchannels of the T-junction PDMS chip were first primed with mineral oil (Sigma, Germany) supplemented with 2% span 80 at the flow rate of 5mL h⁻¹. The cell-laden polymer precursor solution was heated to be around 37°C by a heating module, injected and sheared into monodisperse droplets at the merging points of both fluids. The polymer droplets were in situ crosslinked using the lighting module by exposure to visible light at 405 nm at the outlet. The microgel fabrication process was monitored in real-time by a handheld microscopy, which was connected to the computer by USB. It should be noted that the microfluidic platform is assembled within a clean bench and sterilized using ultraviolet light. The flow rates of both phases were controlled by the infusion pumps to adjust the sizes of cell-laden microgels.”

3. A major point they make is comparison of the histology between the LCAs and the tumors they were derived from. While there are many ways to do this in a systematic way (e.g. blinded histology diagnoses of the tumors and LCAs into major lung cancer histologies and then see how they match) – perhaps the most straightforward thing would be to supply as supplemental data the electronic images of the tumors and the LCAs in H&E stains and the readers can then assess how well they match one another.

Response: According to your suggestion, we have added the images of tumors and LCAs in H&E stains as supplemental data.

4. Another comparison they have provided is bulk RNA expression data of the tumor and the LCAs. As part of this they need to simply state how many mRNA comparisons between different tumors and LCAs they have made for the current heat maps do not easily provide this information. One straightforward way would be to take published mRNA signatures or well known markers for different types of differentiation and see how these compare (e.g. lung adenocarcinoma, lung squamous cancers, large cell, neuroendocrine, mixed histologies). Another straightforward approach is to determine how closely tumors and LCAs from the same patient compare, versus how the LCAs compare to all other patient tumor samples. In addition, they may have done this, but if they compare RNA expression data from individual LCAs from the same patient how similar are they – and how many of these comparisons have they made?

Response: Many thanks for these suggestions. The differentially expressed genes shown in the heatmap (Fig. 4a) were selected as follows: for each patient, we identified the differentially expressed genes as the commonly up-regulated or down-regulated genes in tumor and LCA compared to normal by fold change (>1.5 or <-1.5), and then, the differentially genes for all the four patients (LC14, LC28, LC51 & LC52) were combined in the figure (6,577 genes in total). We revised the figure legend and added more details in the Method section (Line 647-654).

We performed this comparison to demonstrate that LCAs and tumors were highly similar in each patient. As stated above, the differentially expressed genes were selected from each patient, we revised the heatmap by adding white gaps to avoid possible misunderstandings. In general, we followed the second straightforward approach. We conducted RNA sequencing in 4 patients and compared the transcriptome of LCA, Tumor, and Normal within each sample, and discovered that tumors and LCAs shared high similarity in expression profiles and tumor associated function based on the common differentially expressed genes and enrichment analysis pathways. The Pearson correlation coefficients (R values marked at the bottom of the figure) in Figure 4a also showed how similar tumor and LCA of each patient were.

Revised Fig. 4 (a) Gene expression heatmap of 6,577 differentially expressed genes (see Methods) in normal tissues, parental tumors and LCAs of four patients (LC14, LC28, LC51 and LC52). N, normal tissue; T, tumor; A, LCAs.

5. They make a big point about the LCAs being more representative than lung cancer organoids (LCOs) developed in U shaped wells and or LCAs compare to LCOs from the same patient. They need to clearly state how many of these comparisons they have done, what was used for the comparison and how similar or different they are. It may be that LCAs compared to LCOs from the same patient are actually pretty similar for a variety of metrics (e.g. mRNA expression, mutations, freeze thaw stability, drug response phenotypes). Thus, even if LCAs may be better (a lot or only slightly) than LCOs derived by less fancy means, for most usage LCOs may be the easier way to go. Throughout the results they say they have made comparisons, it appears these may be only for a few samples, but some kind of summary of all of the comparisons including number tested for each metric and what was found and how different or similar they are would be useful to know.

Response: We appreciate the reviewer's valuable suggestions. We have revised the related texts to more clearly show the numbers and differences of the comparisons between LCAs and co-cultured assembloids in U-shaped wells in revised manuscript (Line193-196). (kindly explaining that we compared our droplet microfluidic-based assembloids with co-cultured assembloids in U-shaped wells).

“In particular, LCAs **of 5 patients** generated by droplet microfluidics demonstrated a **more uniform shape, cell distribution and TME maintenance** than those generated by coculture in U-bottomed ultralow attachment microplates (ULAs), which are often used to generate 3D spheroid tumor models⁴³ (Supplementary Fig. 4f, g)”.

Indeed, LCOs could meet some research needs but limited in some other studies for the lack of TME and intra-batch consistence, such as evaluation of immuno-oncology and drugs that target the tumor microenvironment. Therefore, researchers had to develop more advanced models to solve these limitations. For the comparisons between LCAs and LCOs, as mentioned in the beginning, the main difference is the tumor environments. And the aim of our droplet microfluidic strategies is to solve the limitation of LCOs which lack TME^{13, 14}, and the limitation of traditional co-culture systems that lack the precise control and intra-batch consistence.

Droplet-based microfluidic systems could manipulate microvolume fluid and generate sub-microliter droplets **in a controlled and high-throughput manner**¹⁵. In our manuscript, **even 10 μ L** of cell-laden hydrogel could be manipulated precisely to produce ≥ 200 LCAs with a volume of **0.05 μ L** within **1 min** by droplet microfluidic method (Fig 2j), enabling the drug screening for personalized medicine, especially for microvolume samples. More importantly, our LCAs generated by droplet microfluidics demonstrated a more uniform shape and cell distribution than those generated by co-culture in ULA microplates or by a 3D bioprinting method (Supplementary Fig. 4f, g). In addition, this system could be integrated and automated control, easy to operate and labor-saving. Similar to tumor organoid applications, the maintenance and expansion of LCOs and the corresponding TME cells are the speed limiting factors for LCAs but absolutely necessary.

In addition, we followed the reviewer's suggestion and provided summary of the comparisons including number tested for each metric and what was found and how different or similar in our manuscript.

6. As part of this the scRNAseq data are important. However, as far as I can tell this only done in a very few examples (2 patients I believe). While more scRNAseq data comparisons would increase the value of the manuscript, they need to start out early in the Results section and state how many comparisons were made, and what the results are – both for LCAs and the tumors, and for comparing LCAs with LCOs from the same patient.

Response: Thanks for pointing out this issue. According to your suggestion, we have presented earlier the number of comparisons for LCAs and tumors in the Results section:

“To further investigate cell-type heterogeneity and cell–cell interaction signatures, we performed single-cell RNA sequencing (scRNA-seq) analysis of parental tumors from **2 patients and corresponding LCAs at 1 and 2 weeks.**”

We have conducted bulk RNA sequencing on 4 patient samples (Figure 4). Preliminary analysis has provided insights into the similarity of assembloids to the primary tumor at the transcriptomic level. Meanwhile, we observed similar enriched gene expression of different tumor microenvironment cells in LCAs to their corresponding tumors, indicating the TME cells maintained in LCAs. To further validate the maintenance of primary tumor transcriptomic and microenvironment characteristics in LCAs, we performed single-cell RNA sequencing analysis of parental tumors from 2 patients and corresponding LCAs at 1 and 2 weeks. Additionally, the presence of the TME was further confirmed across multiple samples through various methods, including immunofluorescence and H&E (Fig. 2c, Fig. 3a, c, d, Suppl. Fig. 4c-e, f). The comprehensive use of these multiple approaches allows us to draw a robust conclusion. Notably, a similar methodology has been employed in articles published in Nature with single-cell RNA-seq data of one sample of each group¹⁶.

We did not design a single-cell RNA sequencing for comparison between LCAs and LCOs and the reason is as follows: The purpose of single-cell sequencing is to comprehensively demonstrate that LCAs cultured for different durations exhibit a similar TME to the primary tumors, and that there is interaction between tumor cells and TME cells (Figure 5). However, lung cancer organoids have been demonstrated to lack the TME^{13,14}, and this has been further validated in our study (Supplementary Fig. 2e). Our LCAs were specifically generated to address this issue by assembling LCOs with TME cells (CAFs and TILs) from the same patient.

7. Obviously a major point the authors make is the potential utility of the LCAs for determining ex vivo drug response phenotypes. What we need are LCA vs. LCO comparisons of drug response phenotypes for drugs commonly used in the treatment of lung cancer. While the IC50 and other values could differ between cells grown in the

two methods, what we need to know are the relative responses of different regimens – the key data that would ultimately be used in selecting therapy for individual patients. Thus, let's say there are a few common chemotherapy regimens used for the treatment of non-small cell lung cancer based on platin doublets (e.g. platin + taxane, platin + gemcitabine, platin + pemetrexed) we need to know for LCAs from one patient: what is the reproducibility of drug response phenotype for each regimen; what are the relative ranks of sensitivity and resistance, and quantitatively how big are these differences? Then, we need to compare this to the same analyses in LCOs from the same patients. Do you get the same relative rankings, are the variations similar or different. Finally, if you look at relative rankings and the quantitative differences in response phenotypes between LCAs from different patients, how similar or different are they? This is the exact type of information one would need to use this assay in a clinical trial and from the current presentation it is not clear what the data would be or are.

Response: We appreciate the reviewer's valuable comments on our manuscript that we used to improve our work. To investigate the difference of drug response phenotype between LCAs and LCOs, we performed drug treatment assays in 6 pairs of LCOs-LCAs using carboplatin-based chemotherapy drugs (PC, Pemetrexed + Carboplatin; TC, Taxol + Carboplatin), commonly used in the treatment of lung cancer. As indicated by the dose-response curves and LogIC50 values, both LCAs and LCOs could recapitulate the inter- and intra-patient heterogeneity of responses to the chemotherapies. Both LCAs and LCOs of different patients showed diversities in their responses to the same chemotherapeutic drug, leading to about 100 times differences in IC50 values. On the other hand, the sensitivities of a LCA or LCO line to two chemotherapies can be extremely different. For example, LCA 42 and LCA36 were sensitive to TC but resistant to PC, whereas LCA19 and LCA22 were sensitive to both PC and TC, whereas LCA66 was resistant to both PC and TC (Response Fig. 2a-c).

Compared to LCA groups, most LCOs showed similar drug response trends such as in LCO19 LCO22 LCO42 and LCO66. However, there were significant difference in other cases. For example, LCO36 was sensitive to both PC and TC, whereas LCA36 was resistant to PC. Additionally, responses of most LCAs to the chemotherapeutic drugs were decreased compared with LCOs (Response Fig. 2a-c), consistent with the previous study¹⁶. These differences may be caused by the tumor microenvironments existed in LCAs, which may reflect the real responses of tumors to drugs^{17, 18, 19}.

For the reproducibility of drug response phenotype, we analyzed all data of the drug assays of LCOs and LCAs above, and performed correlation comparisons between any two sets of replications²⁰. LCA group showed better concordance between viability readings of drug response phenotype compared to LCO group with the R² value in linear regression 0.93 vs. 0.82, suggesting good reproducibility in LCA group (Response Fig. 2d).

Taken together, these results emphasized the necessity of the rapid in vitro drug sensitivity test for choosing the most effective chemotherapy for an individual patient.

LCAs with patient-derived micro-environments may be a promising model for its better reproducibility and more realistic responses to drugs.

Response Figure 2a and 2c have been included in our revised manuscript (new Fig. 7a, b). Response Figure 2b is supplemented in Supplementary Fig. 9a. Response Figure 2d is new Supplementary Fig. 9d. We also revised the relevant manuscript text (line 350-362; line 371-374).

Response Figure 2: (a, b) Heterogeneous responses of LCAs and LCOs derived from lung adenocarcinomas of 6 patients to the carboplatin-based chemotherapies. The fitted dose–response curves illustrated the responses of the LCAs and LCOs to pemetrexed + carboplatin (PC) and taxol + cisplatin (TC) (n = 3 biologically independent cells, data are presented as mean ± SEM). (c) The heat map on the right is LogIC50 values of exemplary LCAs and LCOs in (a, b). R means the IC50 is not available, and the samples are resistant to the drugs. (d) Linear correlation fittings between any two sets of parallel experiments of drug testing assays in LCOs and LCAs derived from the 6 patients (3 replications for every drug concentration).

8. They have made comparisons of the assays in the LCAs with patient responses for 8 patients (Supplementary Table 2). In most of these examples the therapy was given as neoadjuvant therapy (where response rates are very high, and in fact nearly all of the patients experienced what the authors scored as a “PR” with just one stable disease. Because of the complex nature of such comparisons and the fact that each patient was

only treated with one regimen it is impossible to know if LCA guided therapy was actually working in this example. To keep this simple, we need to know for each patient what were the LCAs results of various drug/targeted therapy, immune checkpoint blockade response phenotypes tested and how did they compare to one another was one regimen significantly different (more sensitive or resistant than an of the others? Also, if you compare the results between the 8 patients were the ranking of sensitivity and resistance of the different regimens similar or different? As part of this we know from neoadjuvant studies of NSCLC that 60+ plus of tumors will have a partial response or better to chemotherapy combinations, a similar or higher response if they have EGFR mutations to TKI targeted therapy, and a similar high response to immune checkpoint blockade delivered as neoadjuvant therapy. What is important in terms of long term survival is the generation of a pathologic complete response (or near complete response after such neoadjuvant therapy) and here is where testing of LCA or LCO response phenotypes could help select therapy. Obviously, such information will have to come from subsequent clinical trials.

Response: We appreciate the reviewer's valuable comments. Following the reviewer's suggestions, we performed drug screening among LCAs of 10 patients using various drug/targeted therapy, immune checkpoint blockade (including the LCAs of 7 patients except LC37, which was reminded by editor to remove from the manuscript for enrolled in the unpublished trial NCT04158440).

We first demonstrated that the responses of the LCAs to the targeted therapies were correlated to the genetic mutations of the original tumors using 9 samples, three of which harbor EGFR activating mutations (EGFR-M), the other six have the wild-type EGFR(EGFR-W) (Supplementary Table 1). The drug-response curves of these nine specimens were divided into two groups based on the sensitivities to furmonertinib, perfectly accordant to their genetic mutations (Response Fig. 3a; Revised Fig. 7d). We quantified the responses by calculating the area under the dose-response curve (AUC) and the relative viability at the indicated drug concentration (0.1 μ M), both of which were significantly different between the EGFR-M and the EGFR-W groups ($P=0.0038$ and $P < 0.0001$, respectively) (Response Fig. 3b). The interpolation of the drug-response curves gave half-maximal inhibitory concentration (IC₅₀) values $< 0.07 \mu$ M in the EGFR-M group, whereas $>1 \mu$ M in EGFR-WT group (Response Fig. 3c).

Of note, LCA46 was generated from the biopsy of an AC patient who carried the EGFR mutation before neoadjuvant therapy. We performed drug screening according to the clinical dosing regimen (Taxol + carboplatin + furmonertinib). The drug sensitivity tests indicated that LCA46 were more sensitive Taxol + carboplatin + furmonertinib than to single-agent and double-agent chemotherapy, in agreement with the clinic outcomes (Revised Fig. 7e-g). We also compared the responses to another ALK targeted drug (ensartinib), between LCA63, which harbors the ALK positive expression,

and the other two ALK-negative LCAs (LCA55 and LCA66). LCA63 was more sensitive to ensartinib than the other two LCAs, indicated by the lower drug sensitivity curve and the >10 times smaller IC50 value (Response Fig. 3d, Supplementary Fig. 10g). These results above suggest that the LCAs could predict the patient responses to anticancer targeted therapies.

In addition, from the drug–response curves of the combination chemotherapies, immune checkpoint blockade (PD-1) and immunotherapy combined with chemotherapy, which are commonly used treatment regimens in clinic, we could observe different response phenotypes to the same regimen among LCAs of different patients, and LCAs of the same patient showed different responses to various drug regimens (Response Fig.3e-i). This indicated that LCAs could reflect the individualized differences and selectivity for drug response.

What’s more, we pooled the LogIC50 of all drug responses of the samples interpolated from the fitted dose–response curves, and found that the response rate of combination chemotherapy is significantly higher than that of single-agent chemotherapy (8 of 16 were resistant to single-agent chemotherapies Vs. 5 of 19 resistant to two-agent chemotherapies, 47% VS. 74%)(Response Fig. 3j), consistent with the findings in clinic that doublet chemotherapy improved the overall response rate^{21, 22}. Additionally, for LCAs sensitive to combination chemotherapy, PD-1 plus platinum-based doublet chemotherapy could significantly improve the responses of LCAs derived from different patients (LCA36, LCA66 and LCA69) compared to platinum-based doublet chemotherapy group and PD-1 group (Response Fig. 3j; Supplementary Fig. 10h, i). This was consistent with the previous findings demonstrated that neoadjuvant PD-1 plus chemotherapy improved the event-free survival versus chemotherapy alone in patients with resectable non-small-cell lung cancer^{23, 24, 25}.

Although these drug response results suggested that LCAs might be a promising model for predicting the clinical outcomes, we have to acknowledge that the current sample size is not sufficient to draw a convincing conclusion. We will validate it in more LCAs derived from lung cancer patients and this issue has also been discussed in the Discussion Section (line 489-492) as follows:

“Even though, we have to acknowledge that the current sample size is not sufficient. We will further validate the accuracy of LCA model in predicting clinical drug responses in more samples derived from tumor biopsies before neoadjuvant therapy in the future.”

Response Figure 3a and 3b have been included in our revised manuscript (new Fig. 7d). Response Figure 3c and 3d are new Supplementary Fig. 10a and 10g. Response Figure 3e-i is new Supplementary Fig. 10h. Response Figure 3j is new Supplementary Fig. 10i. We also revised the relevant manuscript text (line 379-422).

Response Figure 3. (a) Responses of LCAs to the EGF inhibitor, furmonertinib, in agreement with EGFR mutations. The fitted dose–response curves represent the viabilities of LCAs of nine different patients exposed to a concentration gradient of furmonertinib ($n = 3$ biologically independent cells, data are presented as mean \pm SEM). Six of the LCAs have wild-type EGFR and the other three harbor EGFR activation mutations. (b) The area under the dose–response curve (AUC-DRC) and the viability at the indicated concentration (0.1 μM) between the EGFR mutation (EGFR-M) and the EGFR wild-type (EGFR-W) groups were compared using the unpaired two-tail Student’s t test. (c) The IC50 values listed in the table were interpolated from the fitted dose–response curves. R means IC50 is not available since the viability of the LCO is $>50\%$ under all the concentrations. (d) The fitted dose–response curves of LCAs of different patients exposed to ALK-targeting drug ensartinib. LCA63 has positive ALK expression, whereas LCA55 and LCA66 have ALK negative expression. (e-i) The fitted dose–response curves of LCAs of different patients exposed to carboplatin-based

chemotherapy drugs, immune checkpoint blockade (PD-1) and PD-1 combined with carboplatin-based chemotherapy drugs. (j) Heat map illustrating the sensitivities of 10 LCA lines to chemotherapies, targeted therapies and PD-1 combined with chemotherapy. The sensitivity is denoted by the $\log_{10}(\text{IC}_{50}(\mu\text{M}))$ value.

Reference

1. Neal JT, *et al.* Organoid Modeling of the Tumor Immune Microenvironment. *Cell* **175**, 1972-1988 e1916 (2018).
2. Naruse M, Ishigamori R, Ochiai M, Ochiai A, Imai T. Gene expression profiles in CAFs exhibited individual variations by the co-culture with CRC organoids. *Cancer Science* **113**, 345-345 (2022).
3. Liu JY, *et al.* Cancer-Associated Fibroblasts Provide a Stromal Niche for Liver Cancer Organoids That Confers Trophic Effects and Therapy Resistance. *Cell Mol Gastroenter* **11**, 407-431 (2021).
4. Cattaneo CM, *et al.* Tumor organoid-T-cell coculture systems. *Nature Protocols* **15**, 15-39 (2020).
5. Zhang YM, *et al.* 3D Bioprinted GelMA-Nanoclay Hydrogels Induce Colorectal Cancer Stem Cells Through Activating Wnt/beta-Catenin Signaling. *Small*, (2022).
6. de la Pena DO, *et al.* Bioengineered 3D models of human pancreatic cancer recapitulate in vivo tumour biology. *Nature Communications* **12**, (2021).
7. Blanco-Fernandez B, Gaspar VM, Engel E, Mano JF. Proteinaceous Hydrogels for Bioengineering Advanced 3D Tumor Models. *Adv Sci* **8**, (2021).
8. LeSavage BL, Suhar RA, Broguiere N, Lutolf MP, Heilshorn SC. Next-generation cancer organoids. *Nat Mater* **21**, 143-159 (2022).
9. Voabil P, *et al.* An ex vivo tumor fragment platform to dissect response to PD-1 blockade in cancer. *Nat Med* **27**, 1250-1261 (2021).
10. Jacob F, *et al.* A Patient-Derived Glioblastoma Organoid Model and Biobank Recapitulates Inter- and Intra-tumoral Heterogeneity. *Cell* **180**, 188-204 e122 (2020).
11. Drost J, Clevers H. Organoids in cancer research. *Nat Rev Cancer* **18**, 407-418 (2018).
12. Fang YC, *et al.* 3D Printing of Cell-Laden Microgel-Based Biphasic Bioink with Heterogeneous Microenvironment for Biomedical Applications. *Adv Funct Mater* **32**, (2022).
13. Hu Y, *et al.* Lung cancer organoids analyzed on microwell arrays predict drug responses of patients within a week. *Nature Communications* **12**, (2021).

14. Kim M, *et al.* Patient-derived lung cancer organoids as in vitro cancer models for therapeutic screening. *Nature Communications* **10**, (2019).
15. Moragues T, *et al.* Droplet-based microfluidics. *Nature Reviews Methods Primers* **3**, (2023).
16. Kim E, *et al.* Creation of bladder assembloids mimicking tissue regeneration and cancer. *Nature* **588**, 664-+ (2020).
17. Ham IH, Lee D, Hur H. Cancer-Associated Fibroblast-Induced Resistance to Chemotherapy and Radiotherapy in Gastrointestinal Cancers. *Cancers* **13**, (2021).
18. Feng B, Wu JZ, Shen B, Jiang F, Feng JF. Cancer-associated fibroblasts and resistance to anticancer therapies: status, mechanisms, and countermeasures. *Cancer Cell Int* **22**, (2022).
19. Saw PE, Chen JN, Song EW. Targeting CAFs to overcome anticancer therapeutic resistance. *Trends Cancer* **8**, 527-555 (2022).
20. Yin SY, *et al.* Patient-derived tumor-like cell clusters for drug testing in cancer therapy. *Sci Transl Med* **12**, (2020).
21. Yi YJ, *et al.* Comparison between single-agent and combination chemotherapy as second-line treatment for advanced non-small cell lung cancer: a multi-institutional retrospective analysis. *Cancer Chemoth Pharm* **86**, 65-74 (2020).
22. Luo L, *et al.* Comparing single-agent with doublet chemotherapy in first-line treatment of advanced non-small cell lung cancer with performance status 2: A meta-analysis. *Asia-Pac J Clin Onco* **11**, 253-261 (2015).
23. Wang C, *et al.* Neoadjuvant nivolumab plus chemotherapy versus chemotherapy for resectable NSCLC: subpopulation analysis of Chinese patients in CheckMate 816. *Esmo Open* **8**, (2023).
24. Forde PM, *et al.* Neoadjuvant Nivolumab plus Chemotherapy in Resectable Lung Cancer. *New Engl J Med* **386**, 1973-1985 (2022).
25. Wang CL, *et al.* Neoadjuvant nivolumab (NIVO) plus chemotherapy (chemo) vs chemo in Chinese patients (pts) with resectable NSCLC in CheckMate 816. *Cancer Res* **83**, (2023).

REVIEWERS' COMMENTS

Reviewer #3 (Remarks to the Author):

NCOMMS-22-46035C

"A patient-specific lung cancer assembloid model with heterogeneous tumor microenvironments."

The authors have made extensive edits and provided additional data requested by the different reviewers in their resubmission of this manuscript all of which is well detailed in their rebuttal section. In going over all of this I realize that there are several obvious things that they need to clarify in their Abstract and Discussion sections.

1. While they have provided some information on how to construct their in house microfluidic device – for anyone to actually try to replicate this would be next to impossible without additional information and advice. I think one way to deal with this in the methods or supplementary data is to provide a statement of the sort – "Detailed instruction and sources of components and materials will be supplied upon reasonable request." A major claim of this report is their claim of a methodological advance in creating assembloids that their in house device provides – since "assembloids" and that approach have already been reported in other contexts as stated in the papers referenced by the authors. Thus, a major reason to publish this paper is that a microfluidics approach similar or identical to that described the authors is one of their main claims.
2. The way the current manuscript is structured in both the Abstract and Discussion is that lung cancer assembloids (LCAs) are "better" than lung cancer organoid (LCO) approach to use for drug testing in clinical translation. In fact, what they have done is using LCO methodology to generate specimens from their initial clinical samples and then feed these along with the tumor microenvironment (TME) into their LCA generation and distribution. It seems to me that they should essentially state this in the Abstract and Discussion sections.
3. The actual comparisons of LCA with LCO on many levels that they provide show that they are very similar with perhaps a slight edge in reproducibility to LCA vs. LCOs for some issues. Again, I think they should state this. As part of this they have some side by side comparisons for LCA and LCO's which is distributed across several portions of figures in the main text and supplementary sections. I really think a figure that pulls all of these comparisons together in one place would really benefit the manuscript and their presentation. Again, I had to dig through all of their various presentations and figures to try to sort this out and most readers will give up trying to do this since it will take a lot of resources to gear up to shift from LCO alone approach to the LCA approach -so why not make it easy to "convince" other labs that this is important to do.
4. At the end of the day the major issue will be prediction of drug, targeted therapy, and immune checkpoint blockade (ICB) therapy responses for individual patients and how these predict or do not predict what will happen in patients. They do not yet have ICB response data that I could find – so first they should state that this information will be vital to obtain going forward. Next, the reproducibility of chemotherapy response phenotypes for LCA and LCO both seem to be very high, with a slight edge to LCA reproducibility. They should state this and indicate that given the small numbers of comparisons made so far this dataset needs to be extended to larger numbers for both LCAs and LCOs.
5. In Figure 7 the key drug response phenotype data are given. First they state "PC" 7B is pemetrexed and carboplatin. However, there is no indication of what "TC" means – I presume this means a taxane and carboplatin – but they really need to have this in the figure legend. Next, they provide a LCA vs. LCO response phenotype comparison of these two platin doublets for 5 patient tumor specimens. From what is presented two of the specimens LC19, LC22 are sensitive to both doublets, one is resistant to both (LC66). There appear to be major discrepancies in response phenotypes for one of the 5 (LC36) – where LCA would select TC over PC while there was no difference (both sensitive) in the LCO from that patient. While there are some quantitative differences for the other between PC and TC they are relatively small. In fact only one tumor LCA22 appears to be really sensitive to platin doublets, and it is to both. So a real possibility is that only LCA22 would really have a good tumor response in the patient. In any event, the authors, need to indicate that more testing is needed.
6. In 7d figure legend it would be useful to know which of the tumor specimens had an EGFR mutation (I presume it is the top 3, LCA19, LCA46, LCA69) but it is really annoying to have to search around to find that out. Of interest, it would appear that even the EGFR mutant tumors

only loss 60% viability with 0.1 micromolar furmonertinib treatment and the difference in IC50s between mutant and wildtype EGFR tumors appears to be only 10-20 fold. It would have been very nice to see what LCO assays resulted in for EGFR targeted therapy. My guess is that they would be very similar to the LCA results – if the authors have this data they should provide it.

7. I would also include in the supplementary section the comparison of tumor and LCA images they provided for Reviewer #3, which the sophisticated reader will want to see.

RESPONSE TO REVIEWERS' COMMENTS

We would like to thank the reviewer for your helpful comments and constructive suggestions, which has significantly improved the presentation of our manuscript. Here we present a detailed point-by-point response to the comments raised by the reviewer.

Reviewer #3 (Remarks to the Author):

The authors have made extensive edits and provided additional data requested by the different reviewers in their resubmission of this manuscript all of which is well detailed in their rebuttal section. In going over all of this I realize that there are several obvious things that they need to clarify **in their Abstract and Discussion sections.**

1. While they have provided some information on how to construct their in house microfluidic device – for anyone to actually try to replicate this would be next to impossible without additional information and advice. I think one way to deal with this in the methods or supplementary data is to provide a statement of the sort – “Detailed instruction and sources of components and materials will be supplied upon reasonable request.” A major claim of this report is their claim of a methodological advance in creating assembloids that their in house device provides – since “assembloids” and that approach have already been reported in other contexts as stated in the papers referenced by the authors. Thus, a major reason to publish this paper is that a microfluidics approach similar or identical to that described the authors is one of their main claims.

Response: We appreciate the reviewer’s valuable suggestion. The statement has been included in the “Methods-Microfluidic Platform Setup” section as following (Line565-566):

“Detailed instruction and sources of components and materials will be supplied upon reasonable request.”

2. The way the current manuscript is structured in both the Abstract and Discussion is that lung cancer assembloids (LCAs) are “better” than lung cancer organoid (LCO) approach to use for drug testing in clinical translation. In fact, what they have done is using LCO methodology to generate specimens from their initial clinical samples and then feed these along with the tumor microenvironment (TME) into their LCA generation and distribution. It seems to me that they should essentially state this in the Abstract and Discussion sections.

Response: The information has been included in the Abstract and Discussion sections as following:

1) Abstract: (Line 35-38)

“This method enables precise manipulation of clinical microsamples and rapid generation of LCAs with good intra-batch consistency in size and cell composition by evenly encapsulating patient tumor-derived TME cells and lung cancer organoids inside microgels. ”

2) Discussion section: (Line 450-453)

“In this study, we generated stable LCOs and corresponding TME cells from patient tumors and then precisely fabricated LCAs by evenly encapsulating TME cells and LCOs inside GelMA-Matrigel microgels with a novel droplet microfluidics-based method.”

3. The actual comparisons of LCA with LCO on many levels that they provide show that they are very similar with perhaps a slight edge in reproducibility to LCA vs. LCOs for some issues. Again, I think they should state this. As part of this they have some side by side comparisons for LCA and LCO's which is distributed across several portions of figures in the main text and supplementary sections. I really think a figure that pulls all of these comparisons together in one place would really benefit the manuscript and their presentation. Again, I had to dig through all of their various presentations and figures to try to sort this out and most readers will give up trying to do this since it will take a lot of resources to gear up to shift from LCO alone approach to the LCA approach -so why not make it easy to “convince” other labs that this is important to do.

Response: We have put the main comparisons between LCOs and LCAs into Supplementary Table 1.

Supplementary Table 1. Comparison of LCOs and LCAs

	LCOs	LCAs																																																								
Morphology																																																										
TME maintenance	DAPI/EpCAM/ α -SMA 	DAPI/EpCAM/ α -SMA 																																																								
Reproducibility of drug testing (Mean Pearson correlation)	0.74	0.89																																																								
Drug responses	    LCOs LCAs 2 1 0 -1 -2   PC TC PC TC     LC19 0.66 -2.23 1.21 0.68    LC22 0.19 -0.75 0.43 -0.07    LC36 -0.96 -0.95 R 0.10    LC42 R -1.03 R 0.88    LC66 R R R R     Log IC50(μM)		LCOs		LCAs		2 1 0 -1 -2	PC	TC	PC	TC	LC19	0.66	-2.23	1.21	0.68		LC22	0.19	-0.75	0.43	-0.07		LC36	-0.96	-0.95	R	0.10		LC42	R	-1.03	R	0.88		LC66	R	R	R	R		Response to Furmonertinib     LCOs LCAs 0.5 0 -0.5 -1.0 -1.5     LC19 -1.6150 -1.1750    LC42 -0.2177 0.6004    LC55 -0.7265 0.0602     Log IC50 (μM)		LCOs	LCAs	0.5 0 -0.5 -1.0 -1.5	LC19	-1.6150	-1.1750		LC42	-0.2177	0.6004		LC55	-0.7265	0.0602	
	LCOs		LCAs		2 1 0 -1 -2																																																					
	PC	TC	PC	TC																																																						
LC19	0.66	-2.23	1.21	0.68																																																						
LC22	0.19	-0.75	0.43	-0.07																																																						
LC36	-0.96	-0.95	R	0.10																																																						
LC42	R	-1.03	R	0.88																																																						
LC66	R	R	R	R																																																						
	LCOs	LCAs	0.5 0 -0.5 -1.0 -1.5																																																							
LC19	-1.6150	-1.1750																																																								
LC42	-0.2177	0.6004																																																								
LC55	-0.7265	0.0602																																																								

Scale bar: 200 μ m; PC, pemetrexed + carboplatin; TC, taxol + carboplatin.

4. At the end of the day the major issue will be prediction of drug, targeted therapy, and immune checkpoint blockade (ICB) therapy responses for individual patients and how these predict or do not predict what will happen in patients. They do not yet have ICB response data that I could find – so first they should state that this information will be vital to obtain going forward. Next, the reproducibility of chemotherapy response phenotypes for LCA and LCO both seem to be very high, with a slight edge to LCA reproducibility. They should state this and indicate that given the small numbers of comparisons made so far this dataset needs to be extended to larger numbers for both LCAs and LCOs.

Response: Thanks for the reviewer suggestions. For the ICB response data, none of the patients received ICB therapy alone, So we could not get the direct prediction of ICB in LCA models. In the future, we will further explore the accuracy of LCAs in assessing ICB treatment in more patients. And we have stated this in the Discussion section as following:

“Even though, we have to acknowledge that the current sample size is not sufficient. We will further validate the accuracy of LCA model in predicting clinical drug responses especially for immune checkpoint blockade therapy in more samples in the future.” (Line 493-496)

We have also indicated that the reproducibility comparison between LCOs and LCAs should be performed in larger numbers of samples in Discussion section as following:

“Our LCA model demonstrated higher consistency across the wells of parallel experiments than LCO model, while the reproducibility of drug response phenotypes for LCA and LCO both seem to be high. Additionally, LCAs and LCOs showed different responses to chemotherapeutic and targeted drugs for the presence of tumor micro-environments. And these comparisons between LCOs and LCAs need be further performed in larger numbers of samples in the future.” (Line 485-491)

5. In Figure 7 the key drug response phenotype data are given. First they state “PC” 7B is pemetrexed and carboplatin. However, there is no indication of what “TC” means – I presume this means a taxane and carboplatin – but they really need to have this in the figure legend. Next, they provide a LCA vs. LCO response phenotype comparison of these two platin doublets for 5 patient tumor specimens. From what is presented two of the specimens LC19, LC22 are sensitive to both doublets, one is resistant to both (LC66). There appear to be major discrepancies in response phenotypes for one of the 5 (LC36) – where LCA would select TC over PC while there was no difference (both sensitive) in the LCO from that patient. While there are some quantitative differences for the other between PC and TC they are relatively small. In fact only one tumor LCA22 appears to be really sensitive to platin doublets, and it is to both. So a real possibility is that only LCA22 would really have a good tumor response in the patient. In any event, the authors, need to indicate that more testing is needed.

Response: We have added TC (taxol + carboplatin) in the figure legend.

In addition, we have indicated that the comparisons of drug responses between LCAs and LCOs should be further performed in larger numbers of samples in the future in Discussion section as following:

“Our LCA model demonstrated higher consistency across the wells of parallel experiments than LCO model, while the reproducibility of drug response phenotypes for LCA and LCO both seem to be high. Additionally, LCAs and LCOs showed different responses to chemotherapeutic and targeted drugs for the presence of tumor micro-environments. And these comparisons between LCOs and LCAs need be further performed in larger numbers of samples in the future.” (Line 485-491)

6. In 7d figure legend it would be useful to know which of the tumor specimens had an EGFR mutation (I presume it is the top 3, LCA19, LCA46, LCA69) but it is really annoying to have to search around to find that out. Of interest, it would appear that even the EGFR mutant tumors only loss 60% viability with 0.1 micromolar furmonertinib treatment and the difference in IC50s between mutant and wildtype EGFR tumors appears to be only 10-20 fold. It would have been very nice to see what LCO assays

resulted in for EGFR targeted therapy. My guess is that they would be very similar to the LCA results – if the authors have this data they should provide it.

Response: The samples with EGFR mutation have been included in the figure legend.

In addition, we have added the only three LCO-LCA pairs of data for the comparisons of response to EGFR targeted drug in Supplementary Figure 9b. LCOs showed a similar response trend to LCAs between *EGFR*-WT and *EGFR*-mutant samples as the reviewer guessed. However, the IC50 values of LCOs groups were significant less (about 3-6 folds) than that of LCAs, indicating that the presence of TME cells could significantly affect the EGFR targeted drug response of the tumors. We will further confirm it in larger numbers of samples in the future work.

Supplementary Figure 9b. Comparison of responses to EGFR targeted drug (Furmonertinib) between LCOs and corresponding LCAs derived from 3 patients. (LC19, EGFR-mutant patient, LC42 and LC55 had no EGFR mutation) (n = 3 biologically independent samples).

7. I would also include in the supplementary section the comparison of tumor and LCA images they provided for Reviewer #3, which the sophisticated reader will want to see.

Response: Thanks for your suggestion and we included the comparison of tumor and LCA images in the supplementary section.